# Plasticity of the *Mycobacterium tuberculosis* respiratory chain and its impact on tuberculosis drug development

Tiago Beites [1,7], Kathryn O'Brien[1,7], Divya Tiwari [1], Curtis A. Engelhart [1], Shaun Walters[1,6], Jenna Andrews[2], Hee-Jeong Yang[2], Michelle L. Sutphen[2], Danielle M. Weiner[2], Emmanuel K. Dayao[2], Matthew Zimmerman[3], Brendan Prideaux[3], Prashant V. Desai[4], Thierry Masquelin[4], Laura E. Via[2,5], Véronique Dartois[3], Helena I. Boshoff [2], Clifton E. Barry III [2,5], Sabine Ehrt [1] & Dirk Schnappinger[1]*

The viability of Mycobacterium tuberculosis (*Mtb*) depends on energy generated by its respiratory chain. Cytochrome *bc1-aa3* oxidase and type-2 NADH dehydrogenase (NDH-2) are respiratory chain components predicted to be essential, and are currently targeted for drug development. Here we demonstrate that an *Mtb* cytochrome *bc1-aa3* oxidase deletion mutant is viable and only partially attenuated in mice. Moreover, treatment of *Mtb*-infected marmosets with a cytochrome *bc1-aa3* oxidase inhibitor controls disease progression and reduces lesion-associated inflammation, but most lesions become cavitary. Deletion of both NDH-2 encoding genes (Δ*ndh-2* mutant) reveals that the essentiality of NDH-2 as shown in standard growth media is due to the presence of fatty acids. The Δ*ndh-2* mutant is only mildly attenuated in mice and not differently susceptible to clofazimine, a drug in clinical use proposed to engage NDH-2. These results demonstrate the intrinsic plasticity of *Mtb*'s respiratory chain, and highlight the challenges associated with targeting the pathogen's respiratory enzymes for tuberculosis drug development.

---

[1] Department of Microbiology and Immunology, Weill Cornell Medical College, New York, NY 10065, USA. [2] Tuberculosis Research Section, Laboratory of Clinical Immunology and Microbiology, NIAID, NIH, Bethesda, MD 20892, USA. [3] Center for Discovery and Innovation, Hackensack Meridian Health, Nutley, NJ 07110, USA. [4] Lilly Research Laboratories, Eli Lilly and Company, Indianapolis, IN 46285, USA. [5] Institute of Infectious Disease and Molecular Medicine, Department of Pathology, University of Cape Town, Cape Town 7925, South Africa. [6]Present address: School of Biomedical Sciences, University of Queensland, Brisbane 4072, Australia. [7]These authors contributed equally: Tiago Beites, Kathryn O'Brien. *email: dis2003@med.cornell.edu

I n 2012, bedaquiline (BDQ) became the first new drug approved for treatment of tuberculosis (TB) in ~40 years[1]. The proposed mode of action of BDQ entails a perturbation of *Mtb*'s ATP synthase causing lethal proton cycling[2]. To function, ATP synthase activity requires proton motive force (pmf) created by the respiratory chain (RC), which in *Mtb* consists of nine respiratory dehydrogenases and four terminal oxidoreductases[3]. Maintenance of pmf is essential for *Mtb*'s viability in replicative and non-replicative states[4], which highlights *Mtb*'s RC as a promising cell process for drug development. Two RC components were predicted to be essential for optimal growth of *Mtb*, namely the cytochrome $bc_1$-$aa_3$ oxidase, encoded by *ctaB*, *ctaC*, *ctaD*, and *ctaE-qcrCAB*, and a type-2 NADH dehydrogenase (NDH-2) encoded by two genes, *ndh* and *ndhA*[5–7]. Recently, *ndh* was shown to be dispensable for growth of *Mtb*[8], but a mutant devoid of NDH-2 activity - due to deletion of both *ndh* and *ndhA* - was predicted to be not viable[8].

Cytochrome $bc_1$-$aa_3$ oxidase and NDH-2 also seem susceptible to inhibition by drug-like molecules and are being pursued as targets for TB drug development. Several groups have identified imidazopyridine amides that likely target QcrB[9–11]. Most notably, Q203, an imidazopyridine amide, was originally reported to be efficacious in the mouse model of TB[10], and it is currently being evaluated in clinical trials (ClinicalTrials.gov NCT02530710; NCT02858973; NCT03563599). Most recently, the arylvinylpiperazine amides were described as a new class of potent QcrB inhibitors[12]. NDH-2 has been proposed as the molecular target of several potent antimycobacterial compounds including the phenothiazines[13] and is thought to activate clofazimine (CFZ)[14], a drug that has been used to treat leprosy for decades and is occasionally included in the treatment of multi-drug resistant (MDR) TB. Other compounds with antimycobacterial activity that have been reported to inhibit NDH-2 include quinolinyl pyrimidines[15], compounds with a tetrahydroindazole or a thioquinazoline core[16], and 2-mercapto-quinazolinones[17]. Furthermore, it has been shown that quinolinequinones are bactericidal due to an activation of NDH-2 activity[18]. This is consistent with a report indicating that activation of the respiratory chain can accelerate the cidality of TB drugs[19].

It is, nevertheless, not obvious why *Mtb* should require either cytochrome $bc_1$-$aa_3$ oxidase or NDH-2 to grow. The *Mtb* genome encodes an alternative, oxygen-dependent respiratory oxidoreductase, the cytochrome *bd* oxidase, which can compensate for the loss of the cytochrome $bc_1$-$aa_3$ in *M. smegmatis*[20] and *Mtb*, suggesting that Q203 might not be able to prevent growth of *Mtb* in mice, unless cytochrome *bd* oxidase is also inhibited[21]. It was furthermore shown that electrons can be re-routed by *Mtb* to

cytochrome *bd* oxidase when cytochrome $bc_1$-$aa_3$ oxidase is inhibited[22]. Cytochrome *bd* oxidase is strongly upregulated in hypoxia suggesting that essentiality of either system may be dependent upon oxygen tension[23,24]. To integrate the somewhat contradictory data on essentiality of cytochrome $bc_1$-$aa_3$ oxidase, it has been proposed that cytochrome *bd* oxidase is sufficient to maintain cell viability when cytochrome $bc1$-$aa_3$ oxidase is not functional, but it is incapable of sustaining growth of *Mtb*[21]. The predicted essentiality of NDH-2 activity is surprising because *Mtb* can express a type-1 NADH dehydrogenase (NDH-1), which is similar to the eukaryotic respiratory complex 1 and thus could, at least in principle, compensate for the loss of NDH-2 activity[3].

Here, we combine conditional gene silencing, chemical genetics, and treatment efficacy studies to determine the consequences of inhibiting *Mtb*'s cytochrome $bc_1$-$aa_3$ oxidase or NDH-2 in vitro and during infections of mice and marmosets.

## Results

**The cytochrome $bc_1$-$aa_3$ and *bd* oxidases of *Mtb* are redundant.** We constructed three TetOFF mutants, in which anhydrotetracycline (atc) represses expression of *ctaE-qcrCAB*, *ctaC*, and *ctaD*, respectively. Atc reduced the growth rate of these mutants, but it did not prevent their growth (Supplementary Fig. 1A–C), supporting the hypothesis that inactivation of the $bc_1$-$aa_3$ complex might not prevent growth of *Mtb*. This conclusion was confirmed by isolating deletion mutants for *ctaE-qcrCAB*, *ctaC*, and *ctaD*. These mutants grew with rates similar to those observed after silencing cytochrome $bc_1$-$aa_3$ subunits by the TetOFF system (Supplementary Fig. 1D–F). Deletion of the targeted genes was confirmed by whole genome sequencing (WGS), which also allowed the screening for possible secondary mutations (Supplementary Fig. 2, and Supplementary Table 1). None of the mutants carried additional mutations predicted to impact growth and genetic complementation rescued the slow growth phenotype of all deletion mutants.

In mice Δ*ctaE-qcrCAB* grew slowly during the acute phase of infection, but by day 28 post infection the strain had reached the same titer as wild-type *Mtb* (WT) and showed no pronounced persistence defect thereafter (Fig. 1). Δ*ctaC* presented an even slower growth rate than Δ*ctaE-qcrCAB*, reaching maximum titers in the lung at day 56 (~1 log lower than the WT), and in spleens at day 112 (~1 log lower than the WT) (Supplementary Fig. 3).

To understand the mechanisms that allow *Mtb* to grow in the absence of cytochrome $bc_1$-$aa_3$ oxidase, we analyzed the mRNA levels of *nuoA* (NDH-1), *ndh* and *ndhA* (NDH-2), *ctaC* (cytochrome $aa_3$ oxidase), *qcrA* (cytochrome $bc_1$ reductase), *cydA*

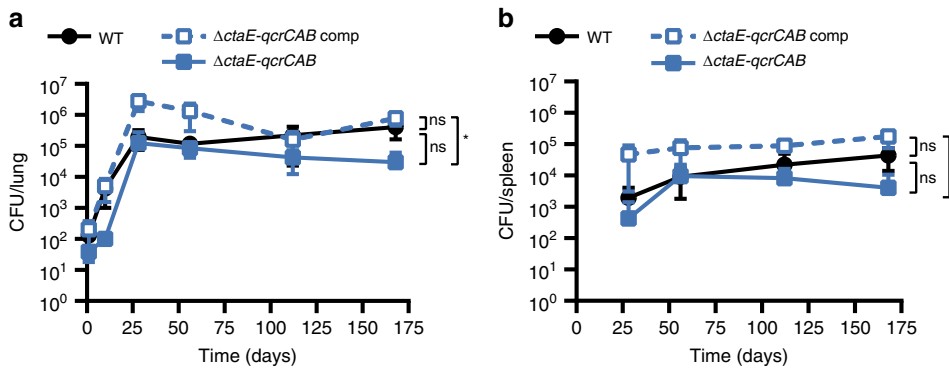

**Fig. 1** The cytochrome bc1-aa3 complex is dispensable for growth and persistence of Mtb in mice. Growth and persistence of Δ*ctaE-qcrCAB* in mouse lungs (**a**) and spleens (**b**). Data are averages of CFUs from at least three mice per time point. Error bars correspond to standard deviation. "Comp" stands for complemented. Statistical significance was assessed by one-way ANOVA followed by post hoc test (Tukey test; GraphPad Prism). *P < 0.05; ns - not significant. Source data are provided as a Source Data file

(cytochrome *bd* oxidase), and *atpE* (ATP synthase) from cultures grown to mid-exponential phase in standard media (Supplementary Fig. 4A–D). *Ndh* and *cydA* were both consistently upregulated in response to genetic inactivation of cytochrome $bc_1$-$aa_3$ oxidase. Induction of *cydA*, which also occurred in response to treatment of *Mtb* with Q203[11], suggested that elevated expression of the cytochrome *bd* oxidase compensated for the lack of cytochrome *bc1-aa3* oxidase. We therefore constructed *Mtb* Δ*cydABDC ctaE-qcrCAB*-TetOFF which allowed us to conditionally silence *ctaE-qcrCAB* in a cytochrome *bd* oxidase deletion mutant (Supplementary Fig. 5). Δ*cydABDC ctaE-qcrCAB*-TetOFF did not grow after silencing of *ctaE-qcrCAB* with atc in vitro (Supplementary Fig. 1G). Doxycycline (doxy), which is used to regulate *Mtb* gene expression during animal infection, had a drastic impact on this strain in mice (Supplementary Fig. 6A, B). When doxy was administered from day 1, the strain was unable to establish infection; when administered from day 35, CFU decreased by approximately five orders of magnitude over the next 35 days. As reported previously[21], we observed no growth or persistence defects for Δ*cydABDC*, in which only the cytochrome *bd* oxidase was inactivated (Supplementary Fig. 1H, Supplementary Fig. 6C, D). In summary, these data demonstrate that *Mtb* requires the cytochrome $bc_1$-$aa_3$ supercomplex to achieve optimal growth rates and maximal titers in mice, but this RC complex is not essential for growth or persistence as long as *Mtb* can express cytochrome *bd* oxidase.

**ND-10885 controlled infection, but increased cavitation**. The findings described above raised concerns about the use of cytochrome $bc_1$-$aa_3$ oxidase inhibitors as stand-alone drugs to treat TB, adding to a previous report showing that Q203 alone was not able to stop *Mtb*'s growth in mice[21]. It is, however, worth noting that Q203 might synergize in vivo with approved TB drugs, such as rifampicin or pyrazinamide[25], thus being a possible suitable candidate for combination therapies. We therefore sought to determine efficacy of a cytochrome $bc_1$-$aa_3$ oxidase inhibitor in an animal model that reproduces more aspects of human TB pathogenesis than the mouse model.

TB disease includes a broad spectrum of lesion types that evolve in response to host attempts to limit the infection. These lesion types present *Mtb* with a variety of different carbon sources to be metabolized under a range of oxygen tensions[26]. Hence, we selected a non-human primate model of TB disease in common marmosets[27,28] to test Q203 efficacy. Initial experiments with Q203 failed to produce a formulation that gave acceptable exposure in these monkeys. A structural analog of Q203 (Supplementary Fig. 7) with improved pharmacokinetics and solubility properties (ND-10885)[29], with an MIC of 0.1 μM against CDC1551, the *Mtb* strain used for infection of marmosets, was therefore employed and found to give good exposure in marmosets following oral administration with co-administration of a CYP450 inhibitor, 1-aminobenzotriazole (ABT)[30,31] (Supplementary Figs. 8A and 9). Tissue distribution studies in lung lesion compartments revealed adequate concentrations in all lesions and favorable partitioning in laser-captured necrotic foci and cavity caseum (Supplementary Fig. 8B–D).

Marmosets were infected by low-dose aerosol and disease development was monitored by bi-weekly 2-[18F]-fluoro-2-deoxyglucose (FDG) PET-CT scanning as previously described[28]. After 50–55 days of infection, moderate to large lesions were clearly visible in these animals by PET-CT imaging and treatment with ND-10885 was initiated. Disease progression was arrested by treatment as illustrated in Fig. 2a, which shows the total uptake of FDG for each of the individual lesions in a single animal. Figure 2b shows the total volume of each lesion within the range

of –100 to 200 Hounsfeld Units, which we have previously shown represents the majority of radiodense tissue associated with TB lesions[28]. Figure 2c, d show heat maps of those same two parameters for all five animals illustrating that most lesions within these animals showed reduced FDG uptake, indicating reduced inflammation, while maintaining a consistent volume of hard lesion density.

After 2 months of treatment, animals were humanely killed and individual lesions were dissected for enumeration of bacterial CFU and detailed histology. The CFU and gross pathology results are shown in Fig. 3 in comparison with two other treatment regimens previously reported: the standard care treatment for TB patients isoniazid-rifampicin-pyrazinamide-ethambutol (HRZE), and an earlier 2-year regimen associated with unacceptable relapse rates in TB patients that comprised streptomycin-isoniazid (HS)[28]. CFU and lesion numbers were substantially higher in the animals treated with ND-10885 compared with a two-month treatment with either HRZE, or HS. None of the bacteria recovered from the infected animals were resistant to ND-10885 (Supplementary Table 2). Most strikingly, as illustrated by the shape of the symbols in Fig. 3a, approximately half of lesions were characterized as cavitary on the basis of gross pathology in animals treated with ND-10885 (inset is the comparison of the gross pathology findings by treatment arm for all three treatments). Histological examination of these lesions revealed that nearly all lesions had a well-developed fibrous cuff surrounding a central area of necrosis. Lesions characterized as "caseous-fibrotic" at gross pathology appear to be identical to cavities that had not yet liquefied and drained into the airways. Figure 3b displays the PET/CT findings from one of these animals and shows the development and drainage of the liquefied interior into the airways during treatment with ND-10885. Finally, Fig. 3c shows the histopathology of the cavity from the left superior lung lobe cavity shown in Fig. 3b revealing a large, partially liquefied lesion with areas of caseous necrosis surrounded by a dense fibrotic cuff. This 8-mm lesion contained ~224,000 bacteria and these bacteria were easily observable within the necrotic region of the lesion by acid-fast staining (Fig. 3c, lower panel). Taken together, these data demonstrate that treatment with *Mtb*'s cytochrome $bc_1$-$aa_3$ oxidase inhibitor ND-10885 effectively controlled the infection in that no new lesions appeared and inflammation was reduced, but only a subset of organisms was affected while those present in existing granulomas flourished resulting in exacerbation of disease by increasing cavitation.

**Essentiality of NDH-2 is conditional**. Besides cytochrome $bc_1$-$aa_3$ oxidase, NDH-2 is the second component of *Mtb*'s RC that is considered as a target for TB drug development. However, efforts to utilize its inhibition might be plagued by the same functional redundancy that affects inhibition of cytochrome $bc_1$-$aa_3$ oxidase. The *Mtb* genome can express two NDH-2 enzymes, encoded by *ndh* and *ndhA*, and one NDH-1, encoded by the *nuo* operon[3]. To evaluate the consequences of inhibiting both NDH-2 enzymes we constructed *Mtb ndh-2*-TetOFF, in which *ndh* has been deleted and *ndhA* can be silenced with atc. *ndh-2*-TetOFF did not grow in standard liquid and solid media containing atc (Fig. 4a). Surprisingly, atc had no effect on *ndh-2*-TetOFF in minimal media devoid of fatty acids (Fig. 4b). We confirmed that essentiality of NDH-2 is conditional using fatty-acid-free media by isolating *Mtb* Δ*ndh-2*, in which *ndh* and *ndhA* have both been deleted. Deletion of both NDH-2 encoding genes was confirmed by WGS (Supplementary Fig. 10). Transformation with an intact copy of *ndh* rescued Δ*ndh-2* (see below), which suggests that the polymorphisms identified in Δ*ndh-2* (Supplementary Table 1) are not responsible for the phenotypes of Δ*ndh-2*. Δ*ndh-2* grew like WT

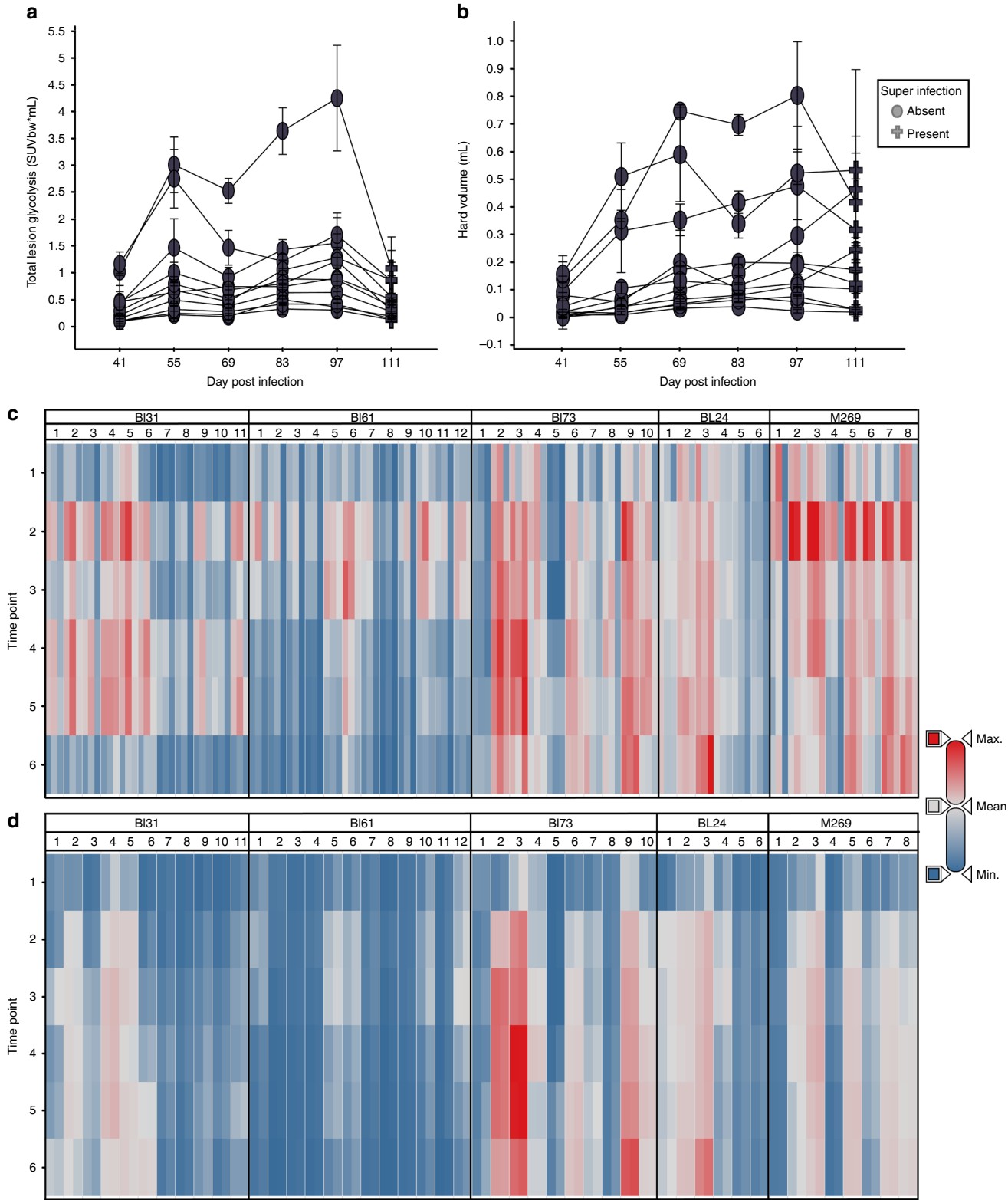

in liquid media without fatty acids, but survival was compromised during gradual oxygen depletion (Fig. 4c). In mouse lungs, Δndh-2 grew to titers that were, on average, 26-fold lower than for WT (Fig. 4d). The mutant was less attenuated in spleens (Fig. 4e).

To assess for possible compensatory effects to the lack of a functional NDH-2 enzyme we analyzed the transcripts levels of nuoA, ndh, ndhA, qcrA, ctaC, cydA, and atpE in cultures grown to mid-exponential phase in fatty-acid-free modified Sauton's

media. Apart from the expected absence of ndh and ndhA transcripts in Δndh-2, we did not observe differences between the mutant and the WT (Supplementary Fig. 4E, F). To test if Δndh-2 is viable due to compensation by NDH-1 we determined susceptibility of Δndh-2 to the NDH-1 inhibitor rotenone. Rotenone was inactive against WT Mtb, which agrees with previous data;[4] but rotenone prevented growth of and was bactericidal for Δndh-2 (Fig. 5a, b). We conclude that NDH-1 and

**Fig. 2** ND10885 treatment of M. tuberculosis infected marmosets. **a** Total lesion glycolysis within each individual lesion for a single animal. The timeline begins 41 days post-infection and treatment within this animal begins at Day 55. Most lesions appear to become less FDG avid within the first 2 weeks of treatment and then stabilize or progress for 2-month treatment period. In untreated animals all lesions would have continued to progress. Symbols represent the mean and error bars of the standard deviation of three independent readers. **b** Hard lesion volume (−100 HU to +200 HU) in most lesions the volume growth slows when treatment starts but it is maintained at the same level as the last pretreatment scan (Day 55). Symbols represent the mean and error bars the standard deviation of three independent readers. **c** Heat map of the lesion-level total PET data for all five animals that were treated with ND10885. Timepoint 2 is the equivalent of Day 55 in panels (**a**) and (**b**), the starting point of drug administration. Most lesions in these animals behave in the same way as those shown in panel (**a**), the total inflammation in each decreases from timepoint 2 to the end of therapy. For each lesion the three bars represent SUVmax, SUVmean and total lesion glycolysis. **d** The lesional volume change in each lesion for all animals across the same treatment period. Again similar to the pattern shown in panel (**b**) volume plateaus in most lesions at timepoint 3, 2 weeks after the initiation of treatment. For each lesion two bars are shown, the left bar indicates volume in the hard range (−100 to + 200 HU) while on the right the total lesion volume (from −500 to + 200 HU) is shown. Source data are provided as a Source Data file

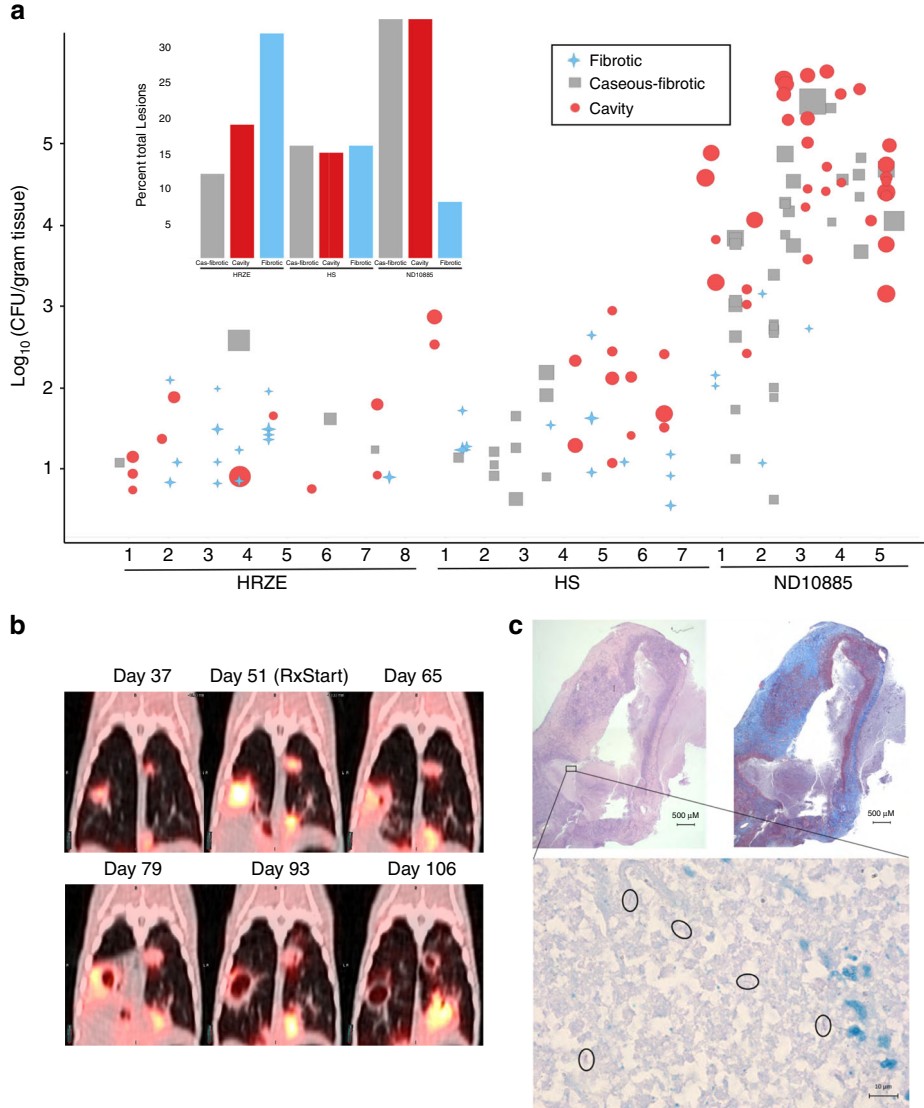

**Fig. 3** ND10885 treatment in infected marmosets allows cavity formation. **a** Lesion pathology at necropsy for marmosets treated with three different drug regimens. HRZE, isoniazid, rifampicin, pyrazinamide, and ethambutol; HS, isoniaizid, and streptomycin (CFU data and PK have been previously published for both control arms; ref. [28]). Each number refers to an individual animal and each symbol above it describes the pathology of each individual lesion within that animal, the size of the symbol is according to the size of the lesion, small symbols are 1 mm lesions and the largest symbols are ~10 mm lesions. The inset bar graph summarizes the same data according to the total percentage of lesions in each of the three pathology types. Source data are provided as a Source Data file. **b** an example of PET/CT images of one animal treated with ND10885 over time, the first image is 37 days after infection with a low-dose aerosol of M. tuberculosis, the second image is day 51 and immediately after that image the animal began receiving ND10885. There are two lesions, one on the right and one on the left that progressively cavitate and expel their contents. **c** Pathology images of one of these cavities showing on the top left the H&E stained slice and on the right the Masson's Trichrome stain to illustrate the fibrous cavity wall. In both panels the scale bar is 500 μm. The small black box in the H&E slice is shown at ×100 with a 10-μm scale bar with acid-fast staining showing the presence of multiple individual bacilli (circled)

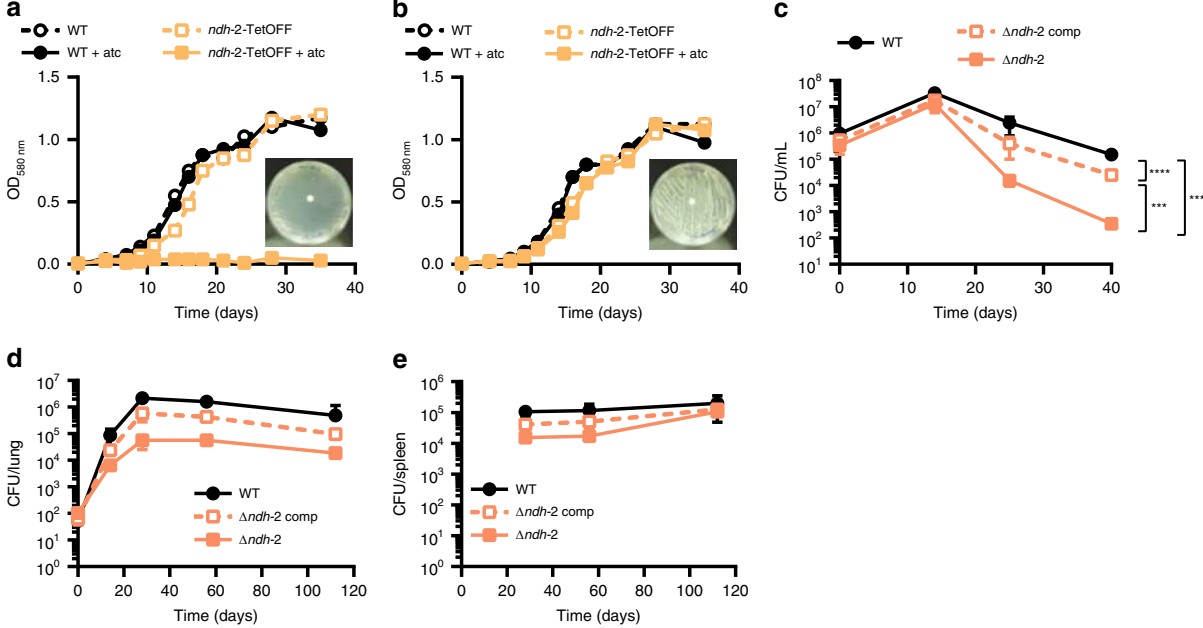

**Fig. 4** NDH-2 is essential for growth with fatty acids. Growth with (**a**) or without fatty acids (**b**). Data are representative of two independent experiments. **c** Growth and survival in the Wayne model of hypoxia. Data are averages of six to eight data points from two independent experiments. Growth and persistence in lungs (**d**) and spleens (**e**) of C57BL/6 mice. Data are averages from four mice per time point and representative of two independent infections. Error bars correspond to standard deviation. "Comp" stands for complemented. Statistical significance was assessed by one-way ANOVA followed by post hoc test (Tukey test; GraphPad Prism). Differences between strains in (**d**) and (**e**) are not significant. ***$P < 0.001$; ****$P < 0.0001$. Source data are provided as a Source Data file

NDH-2 are functionally redundant and inhibition of both kills *Mtb* regardless of the presence of fatty acids.

We next sought to understand why fatty acids were toxic for Δ*ndh-2*. Δ*ndh-2* was hypersusceptible to oleic acid (C18:1), palmitic acid (C16:0), and octanoic acid (C8:0), yet butyric acid showed only mild toxicity and Δ*ndh-2* grew normally with acetate and cholesterol (Fig. 5c, Supplementary Fig. 11). Fatty acid toxicity was thus dependent on chain length, with longer fatty acids being more toxic. This suggested that NDH-2 might be needed to balance the ratio of NADH to NAD$^+$, similar to the function energy dissipating respiratory enzymes have during aerobic growth with highly reduced carbon sources in other bacteria[32,33]. To test this, we expressed LbNox (Supplementary Fig. 12), an NADH oxidase which converts NADH into NAD$^+$ and water[34], in Δ*ndh-2*. LbNox reduced sensitivity of Δ*ndh-2* to oleic acid to the same extent as ectopic expression of Ndh (Fig. 5c). When we measured NADH and NAD$^+$, we found the NADH/NAD$^+$ ratio to be higher in Δ*ndh-2* than in WT *Mtb* (Fig. 5d). Complementation with *ndh* and expression of LbNox reduced the NADH/NAD$^+$ ratio of Δ*ndh-2* to a level similar to that of WT. However, the NADH/NAD$^+$ ratio of Δ*ndh-2* was high with and without fatty acids and oleic acid did not cause NADH to accumulate further (Fig. 5d). The latter argued against fatty acid toxicity being a consequence of an imbalance in NADH/NAD$^+$ in Δ*ndh-2*. We reasoned that accumulation of NADH after exposure to oleic acid was prevented by NDH-1.

In contrast to NDH-2, consumption of NADH by NDH-1 is linked to the export of protons. Sole reliance on NDH-1 might thus cause an excessive accumulation of periplasmatic protons in the presence of fatty acids and increase proton motif force (pmf) to levels that inhibit growth. To test this hypothesis, we used valinomycin or nigericin, which can reduce pmf by abrogating the membrane potential and ΔpH, respectively. A narrow range of valinomycin concentrations indeed supported growth of Δ*ndh-2* with oleic acid. In contrast, no effect was observed with nigericin

or rifampicin (Fig. 5e, f, g). This suggested that a deregulation of the membrane potential and consequently of pmf causes the oleic acid sensitivity of Δ*ndh-2* (Fig. 5h).

**Drug susceptibility profiles**. NDH-2 and cytochrome $bc_1$-$aa_3$ oxidase have been proposed as the targets for several small molecules with antitubercular activity (Fig. 6a). To determine how genetic inactivation of NDH-2 and terminal oxidases affects drug susceptibility of *Mtb* we profiled Δ*ctaE-qcrCAB*, Δ*cydABDC*, and Δ*ndh-2* with (i) the TB drugs rifampicin, isoniazid, ethionamide, and ethambutol; (ii) compounds targeting NDH-1 (rotenone, ROT; piericidin A, PIR; pyridaben, PYR), the ribosome (linezolid), DNA gyrase (ciprofloxacin), ATP synthase (bedaquiline, BDQ), and cytochrome $bc_1$ reductase (Q203); (iii) compounds proposed to target NDH-2 (CFZ, three phenothiazines (chlorpromazine, CPZ; trifluoperazine, TFP; thioridazine, TRZ), and two 2-mercapto-quinazolinones (DDD00853663, DDD00946831, which correspond to compounds 7 and 10 in the work that first described them – Supplementary Fig. 13[17]); and (iv) the ionophore valinomycin. Susceptibility to rifampicin, isoniazid, ethionamide, ethambutol, linezolid, and valinomycin was similar for all strains (Supplementary Tables 3 and 4), demonstrating that none of the mutants had an unspecific drug susceptibility defect. Q203 was more potent against Δ*cydABDC* than WT (Supplementary Fig. 14A), which is in agreement with previous reports[11,21,35]. Q203 showed no activity against Δ*ctaE-qcrCAB* (Fig. 6g). At least in standard media, Q203 has thus no phenotypically relevant target other than cytochrome $bc_1$-$aa_3$ oxidase. Δ*ctaE-qcrCAB*, but not Δ*cydABDC*, proved to be hypersusceptible to NDH-1 inhibitors (Fig. 6h, Supplementary Fig. 14B–D, G, H).

2-mercapto-quinazolinones have recently been described as inhibitors of NDH-2[17]. We found 2-mercapto-quinazolinones to be potent against WT when oleic acid was present in the media,

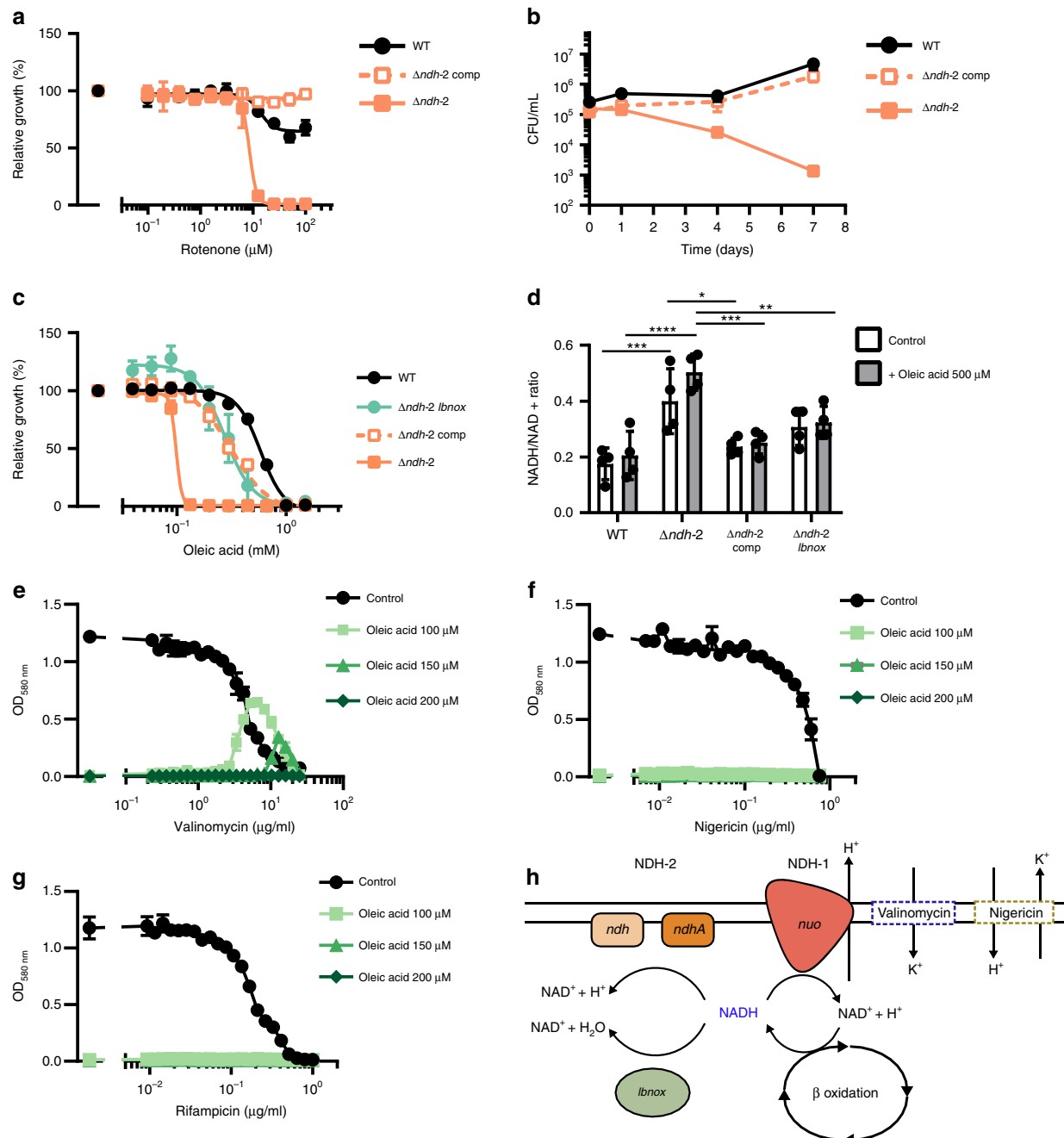

**Fig. 5** NDH-2 is required for redox homeostasis and counteracts inhibition of NDH-1. **a** Impact of rotenone on growth. **b** Survival in fatty-acid-free Sauton's minimal medium supplemented with 50 µg/mL rotenone. **c** Impact of oleic acid on growth. (**d**) NADH/NAD + intracellular ratio after 6 h in modified Sauton's minimal medium supplemented with DMSO (1%, control), or oleic acid (500 µM). Impact of valinomycin (**e**), nigericin (**f**), and rifampicin (**g**) on growth in modified Sauton's minimal medium supplemented with DMSO (1%; control), or oleic acid at different concentrations (100 µM, 150 µM, and 200 µM). **h** Proposed mechanism of NDH-2 essentiality in the presence of fatty acids. Data are averages of three data points and representative of at least two independent experiments. Error bars correspond to standard deviation. "Comp" stands for complemented. Statistical significance was assessed by one-way ANOVA followed by post hoc test (Tukey test; GraphPad Prism). *$P < 0.05$; **$P < 0.01$; ***$P < 0.001$; ****$P < 0.0001$. Source data are provided as a Source Data file

yet inactive against WT and Δ*ndh-2* (Fig. 6F, Supplementary Figs. 14O and 15E, F) in fatty-acid-free media. Chemical and genetic inhibition of NDH-2 thus lead to similar phenotypes. Interestingly, both 2-mercapto-quinazolinones were also more active against Δ*cydABDC* than WT, which suggests that the inactivation of both cytochrome *bd* oxidase and NDH-2 synergize to inhibit *Mtb*'s growth (Supplementary Fig. 14E, F). The effect of

these compounds on Δ*ctaE-qcrCAB* growth was negligible (Supplementary Fig. 14I, J). Δ*ndh-2* was furthermore hypersusceptible to all NDH-1 inhibitors and BDQ (Fig. 6b, c, Supplementary Fig. 14K, L). However, inactivation of NDH-2 had little, if any, effect on the phenothiazines and CFZ (Fig. 6d, e, Supplementary Fig. 14M, N). CFZ and phenothiazines also had similar activities in the presence and absence of oleic acid

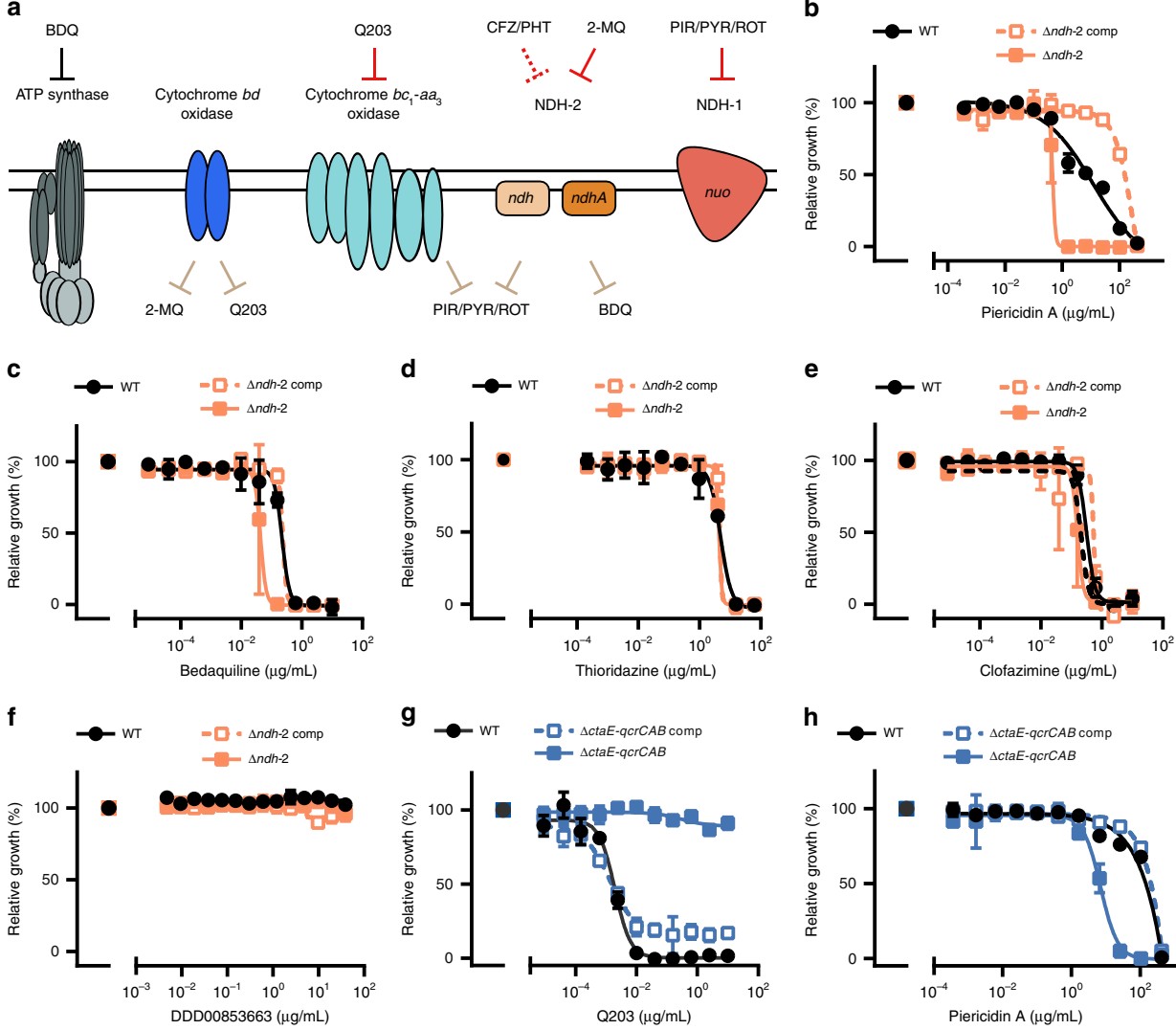

**Fig. 6** Chemical–genetic interactions. **a** Summary of chemical–genetic interactions. Red lines represent direct interactions that are consistent (solid red lines) or inconsistent (dotted red lines) with the chemical–genetic interactions defined in this study. Brown lines represent chemical–genetic interactions that have been defined in this study and are most likely indirect. Impact of piericidin A (**b**, **h**) bedaquiline (**c**) thioridazine (**d**), clofazimine (**e**), DDD00853663 (**f**), and Q203 (**g**) on growth. Data are averages of three cultures and represent at least two independent experiments. Error bars correspond to standard deviation. BDQ, bedaquiline; CFZ, clofazimine; PHT, phenothiazines; PIR, piericidin A; PYR, pyridaben; ROT, rotenone; 2-MQ, 2-mercapto-quinazolinones. "Comp" stands for complemented. Source data are provided as a Source Data file

(Supplementary Fig. 15A–D). We concluded that CFZ does not require activation by NDH-2 to prevent growth of *Mtb* and that NDH-2 is not the primary, growth-limiting target of the phenothiazines.

## Discussion

We draw three main conclusions from this work. First, inactivation of *Mtb*'s cytochrome *bc₁-aa₃* oxidase alone is insufficient to kill *Mtb* and could possibly complicate treatment by increasing cavity formation. Complete genetic inactivation of cytochrome *bc₁-aa₃* oxidase decreased but did not abolish growth of *Mtb* in vitro and in mice. Viability of *Mtb* without cytochrome *bc₁-aa₃* oxidase was dependent on a functional cytochrome *bd* oxidase. In contrast, inactivation of both cytochrome *bc₁-aa₃* oxidase and cytochrome *bd* oxidase rapidly killed *Mtb* in mice. Marmosets provide an animal model that closely recapitulates human disease[27] with frontline chemotherapy following a similar trajectory to sterilization in patients[28]. Strikingly, 2 months of treatment

with the cytochrome *bc₁-aa₃* oxidase inhibitor ND-10885 reduced inflammation in *Mtb*-infected marmosets as measured by FDG uptake but caused no discernable reduction in radiodense volume of disease. ND-10885 arrested disease progression but not pathology progression. We have previously demonstrated that efficacious drugs and drug regimens lead to a progressive decline in radiodensity corresponding to granulomatous regions in the lung[28,36]. Such caseous lesions are severely hypoxic[37] and replication of *Mtb* under these conditions seems unlikely to be dependent upon the cytochrome *bc₁-aa₃* oxidase targeted by ND-10885. Failure to restrict the growth of these hypoxic bacilli likely leads to progression of disease resulting in cavity formation and high bacterial titers. Our data also suggest that growth of bacteria in cellular lesions and macrophages is restricted as lesion density stabilizes as would be expected if in these microenvironments *Mtb* was dependent upon the cytochrome *bc₁-aa₃* oxidase. Correspondingly, ND-10885 had inferior sterilizing ability compared to even sub-therapeutic regimens such as streptomycin/isoniazid. Alarmingly, treatment with ND-10885 led to increased cavitation,

a pathology known to be associated with poor treatment outcome[38]. Whether this increase in cavitation is specific to ND-10885 or will also be caused by other inhibitors of the cytochrome $bc_1$-$aa_3$ oxidase remains to be determined.

Second, preventing growth of Mtb during infection by targeting its NADH dehydrogenases requires inactivating three enzymes: the two NDH-2 enzymes encoded by ndh and ndhA and the NDH-1 complex encoded by the nuo operon. Previous genetic analysis concluded that Mtb requires at least one NDH-2 enzyme to grow[8]. Our data agree with this prediction as we found Mtb to require NDH-2 for growth in standard media. Mtb does, however, not require NDH-2 to grow in fatty-acid-free medium or in media containing short-chain fatty acids or cholesterol. In mice, $\Delta ndh$-2 was attenuated during the acute phase of infection, but it showed no severe persistence defect. The attenuation that results from inactivation of both NDH-2 enzymes is thus not fundamentally different from what has been observed for a mutant missing NDH-1 and one of Mtb's two NDH-2 enzymes. The activity of specific NDH-2 inhibitors, such as the 2-mercapto-quinazolinones, phenocopy genetic inactivation of NDH-2 in vitro and will thus likely be inefficacious during infection.

Complementation of $\Delta ndh$-2 oleic acid hypersusceptibility with LbNox indicated that NADH generation contributes to long chain fatty acid toxicity. This can explain why $\Delta ndh$-2 is less sensitive to short-chain fatty acids and cholesterol, as the full oxidation of these carbon sources require a low number of β-oxidation cycles, while, for example, oleic acid goes through 9 cycles for full oxidation. However, toxicity did not arise from an unbalanced NADH/NAD$^+$ ratio alone. Rescue with valinomycin, but not with nigericin, indicated that a deregulation of the membrane potential was, at least in part, responsible for fatty acid toxicity. Since NDH-1 is the only NADH dehydrogenase operating in $\Delta ndh$-2, the coupling of NADH oxidation with proton extrusion might have contributed to disturb the membrane potential and consequently pmf.

Third, phenothiazines and CFZ inhibit growth of Mtb independently of NDH-2. Although there is evidence for an effect of phenothiazines on Mtb's RC[4,13], the potency of these compounds does not change in response to inactivation of NDH-2. That phenothiazines act independently of NDH-2 is also supported by the pleiotropic effects that phenothiazines can have, which include an impact on membrane permeability and inhibition of the synthesis of various macromolecules[39,40]. The growth-inhibitory activity of CFZ is thought to depend on the oxidative stress it causes after reduction by NDH-2[14]. Antioxidants can rescue Mtb's viability when facing bactericidal CFZ concentrations, which supports this mode of action[14]. However, WT and $\Delta ndh$-2 were equally susceptible to CFZ. Clearly, the activity of CFZ is not dependent on NDH-2. Conceivably, CFZ might be reduced by enzymes other than NDH-2, but it is unlikely that NDH-1 is capable of fulfilling this role, since membranes from bacteria that only express NDH-1 did not reduce CFZ[14].

After the discovery of BDQ, several complexes of Mtb's RC were embraced as new targets for TB drug development. This enthusiasm was supported by new small molecules that inhibit the cytochrome $bc_1$-$aa_3$ oxidase and drugs or drug candidates that engage NDH-2. However, the data presented here suggest that inactivation of either cytochrome $bc_1$-$aa_3$ oxidase or NDH-2 alone may not improve TB chemotherapy. Combinations of inhibitors targeting either cytochrome $bc_1$-$aa_3$ oxidase and cytochrome bd oxidase or NDH-2 and NHD-1 will be required to effectively inhibit the respiratory chain of Mtb. In this context, our work further unveiled genetic and chemical–genetic interactions which could be exploited to improve TB chemotherapy, namely the interactions between cytochrome $bc_1$-$aa_3$ oxidase and NDH-1, NDH-2 and ATP synthase, and NDH-2 and cytochrome

bd oxidase. Targeting only cytochrome $bc_1$-$aa_3$ oxidase or NDH-2 might be most valuable for the treatment of infections caused by mycobacteria, such as M. leprae, that do not possess a functional cytochrome bd oxidase or NDH-1. This strategy has been successfully applied to M. ulcerans, which lacks the alternative terminal oxidase[41].

## Methods

**Culture conditions**. E scherichia coli was used as host for cloning and cultured in LB. Mtb was generally cultured in Middlebrook 7H9 supplemented with 0.2% glycerol, 0.05% tyloxapol, and ADNaCl (0.5% BSA, 0.2% dextrose and 0.85% NaCl) or in Middlebrook 7H10 supplemented with 0.5% glycerol and 10% oleic acid-albumin-dextrose-catalase (OADC). Fatty acid sensitive strains were cultured in a modified Sauton's minimal medium (0.05% potassium dihydrogen phosphate, 0.05% magnesium sulfate heptahydrate, 0.2% citric acid, 0.005% ferric ammonium citrate, and 0.0001% zinc sulfate) supplemented with 0.05% tyloxapol, 0.4% glucose, 0.2% glycerol, and ADNaCl with fatty-acid-free BSA (Roche). Fatty-acid-free solid medium contained 1.5% bactoagar (BD) and glycerol was used at a concentration of 0.5%. When necessary, antibiotics were added at the following concentrations: apramycin 20 μg mL$^{-1}$, kanamycin 25 μg mL$^{-1}$, zeocin 25 μg mL$^{-1}$, streptomycin 25 μg mL$^{-1}$, and hygromycin 50 μg mL$^{-1}$. When needed, atc was used at a concentration of 500 ng mL$^{-1}$ (ref. [42]).

**Mutant construction**. All mutants were constructed using protocols and procedures described in detail elsewhere[43]. Briefly, for mutants of ctaE-qcrCAB, we first generated a merodiploid by inserting a plasmid at the att-L5 site that expressed ctaE-qcrCAB under the control of its own promoter (pGMCS-PctaE-qcrCAB). The WT copy of ctaE-qcrCAB was then replaced by a hygromycin resistance cassette using a temperature sensitive plasmid leading to $\Delta$ctaE-qcrCAB::PctaE-qcrCAB. In this strain, the only functional copy of ctaE-qcrCAB is located in the attL5 site, where it can be replaced or deleted by site-specific recombination[44]. To construct ctaE-qcrCAB-TetOFF, pGMCS-PctaE-qcrCAB was replaced with pGMCZ-T38S38-750-ctaE-qcrCAB-SD2, which expresses under control of a reverse TetR; to delete ctaE-qcrCAB, pGMCS-PctaE-qcrCAB was replaced with pGMCZq17-0×0×, which does not contain ctaE-qcrCAB. To complement $\Delta$ctaE-qcrCAB the mutant was transformed with pGMCtKq1-PctaE-qcrCAB, which expressed ctaE-qcrCAB under the control of its own promoter after integration into the tweety phage attachment site.

To generate ctaC-tetOFF and ctaD-tetOFF, we have used the dual control (DUC) strategy, which combines repression of transcription and controlled proteolysis to deplete the targeted protein[45]. First merodiploids were generated by introducing a copy of ctaC under the control of a strong promoter (pGMCS-P750-ctaC-rv2219) or ctaD-serB2 under the operon's own promoter (pGMCS-PctaD-serB2) into the att-L5 site, the WT loci were replaced by a hygromycin resistance cassette using the phage-mediated recombineering[46]. These strains were transformed with a plasmid expressing sspB (pGMCtZq17-TSC10M1-sspB) that integrated at the tweety phage attachment site. Next, we swapped the construct at att-L5 with a plasmid where ctaC-DAS or ctaD-DAS are under the control of a Tet-OFF system (pGMCZ-T38S38-750-ctaC-DAS; pGMCZ-serB2(tr)-T38S38-750-ctaD-DAS). In the ctaD-TetOFF mutant, serB2 is expressed constitutively. To generate $\Delta$ctaC and $\Delta$ctaD, we used $\Delta$ctaC::pGMCS-P750-ctaC-rv2219 and $\Delta$ctaD:: pGMCS-PctaD-serB2, and swapped the construct at the att-L5 site by an empty plasmid (pGMCZq17-0×0×) in the case of $\Delta$ctaC, and a plasmid expressing serB2 constitutively (pGMCZ-PserB2) in the case of $\Delta$ctaD. Complemented strains were obtained by introducing ctaC under the control of a strong promoter (pGMCtKq1-P750-ctaC), or ctaD under the control of a strong promoter (pGMCtKq1-Puv15-ctaD), at the tweety phage attachment site.

$\Delta$cydABDC was constructed by replacing cydABDC with a hygromycin resistance cassette via homologous recombination. The complemented strain was obtained by expressing cydABDC under its own promoter (pGMCK-PcydABDC-SD1) at the att-L5 attachment site. To construct $\Delta$cydABDC ctaE-qcrCAB-TetOFF, we took advantage of the loxP sites flanking the hygR cassette, which enabled its removal by expression of Cre recombinase, rendering a hygromycin sensitive strain. Using this strain as genetic background, we have followed the strategy previously described to generate a ctaE-qcrCAB-TetOFF strain.

To obtain ndh-2-TetOFF and $\Delta$ndh-2 we first generated a merodiploid that expressed ndh under its native promoter (pMPa-Pndh) at attL5. We then successively replaced the WT copies of ndh and ndhA with hygromycin and kanamycin resistance cassettes through homologous recombination, respectively, giving rise to $\Delta$ndh/ndhA::Pndh. The ndh copy in $\Delta$ndh/ndhA::Pndh was then swapped against a plasmid containing ndhA under the control of the DUC system (pGMCS-0×750-ndhA-DAS)[45]. The resulting strain transformed with the sspB expressing vector (pGMCtZq17-TSC10M1-sspB) to generate ndh-2-TetOFF. $\Delta$ndh-2 was obtained by swapping the Pndh plasmid against an empty vector (pGMCZq17-0×0×). The complemented strain was obtained by introducing ndh under the control of a strong promoter (pGMCtSq19-0×-Puv15-ndh) at the tweety phage attachment site. For complementation with LbNOX, a codon usage adapted gene (lbnox$_{Mtb}$) was ordered from GenScript and cloned into an integrative plasmid (pGMCgS-0×-Ptb38-LbNOX-FLAG-SD1) that inserts at the Giles

attachment site. Strains and plasmids used in this work are listed in Supplementary Tables 5 and 6.

**Whole genome sequencing.** The genetic identity of Δ*ctaE-qcrCAB*, Δ*ctaC*, Δ*ctaD*, and Δ*ndh-2* was confirmed by whole genome sequencing (WGS). Between 150 and 200 ng of genomic DNA was sheared acoustically and HiSeq sequencing libraries were prepared using the KAPA Hyper Prep Kit (Roche). PCR amplification of the libraries was carried out for 10 cycles. $5–10 \times 10^6$ 50-bp paired-end reads were obtained for each sample on an Illumina HiSeq 2500 using the TruSeq SBS Kit v3 (Illumina). Post-run demultiplexing and adapter removal were performed and fastq files were inspected using fastqc (Andrews S. (2010). FastQC: a quality control tool for high throughput sequence data. Available at: http://www.bioinformatics.babraham.ac.uk/projects/fastqc). Trimmed fastq files were then aligned to the reference genome (M. tuberculosis H37RvCO; NZ_CM001515.1) using bwa mem[47]. Bam files were sorted and merged using samtools[48]. Read groups were added and bam files de-duplicated using Picard tools and GATK best-practices were followed for SNP and indel detection[49]. Gene knockouts and cassette insertions were verified for all strains by direct comparison of reads spanning insertion points to plasmid maps and the genome sequence. Reads coverage data was obtained from the software Integrative Genomics Viewer (IGV)[50–52]. Sequencing data was deposited in NCBI's Sequence Read Archive (SRA) database under the BioProject ID PRJNA532433.

**Quantitative PCR.** Cultures were grown until mid-exponential phase ($OD_{580nm}$ of 0.5) and harvested with GTC buffer. RNA was extracted using TRIzol reagent (Invitrogen) and "Quick-RNA™ Miniprep Kit" (Zymo Research), following the manufacturer's instructions. DNA was digested with the "Turbo DNA-free kit" (Invitrogen), and 500 μg of RNA were used to synthesize cDNA using M-MuLV Reverse Transcriptase (New England BioLabs). qPCR reaction was performed with LightCycler® 480 SYBR Green I Master (Roche) in the LightCycler® 480 System (Roche). Primers and probes were designed in the software RealTimeDesign™ (Biosearch Technologies), and ordered from TIB MOLBIOL, LLC. The sequences of all primers and probes are available in Supplementary Table 7. Probes for genes of interest were labeled with 5′ 6-FAM/ 3′ BHQ1, while the probe for the reference gene *sigA* was labeled with 5′ LC670.

**Wayne model of hypoxia.** *Mtb* strains were grown in Dubos broth base (Difco) supplemented with 0.05% tyloxapol, 0.75% dextrose, 0.5% fatty-acid-free BSA (Roche), and 0.85% NaCl until mid-log phase ($OD_{580}$ approx. 0.4). Single cell suspensions were prepared in the same medium to a final $OD_{580}$ of 0.05. In total, 20 mL of the single-cell suspension were aliquoted into flat-bottomed tubes each equipped with a black cap (Wheaton) and a magnetic stirrer. Paraplast wax (Kendall) was used to seal the tubes. One tube of each strain was further supplemented with methylene blue (1.3 μg mL$^{-1}$), which served as a hypoxia indicator. Cultures were incubated at 37 °C on a magnetic platform at 130 rpm. $OD_{580}$ was measured for 40 days and bacteria were plated for CFU at the indicated time points in fatty-acid-free modified Sauton's medium.

**Growth with fatty acids.** Strains were grown in modified Sauton's minimal medium until mid-log phase ($OD_{580}$ 0.6–0.8). Bacteria were collected and resuspended in the same medium to generate single-cell suspensions. Modified Sauton's minimal medium was supplemented with fatty acids or cholesterol at the following final concentrations: acetic acid 125 mM, butyric acid 25 mM, octanoic acid 2 mM, palmitic acid 500 μM, oleic acid 1 mM (1.5 mM in the assays with *lbnox*), and cholesterol 125 μg mL$^{-1}$. Octanoic acid, palmitic acid, and oleic acid were solubilized in DMSO, while cholesterol was solubilized in a 1:1 tyloxapol:ethanol solution and then added to the medium. Serial dilutions of each fatty acid or cholesterol (2-fold) were made in 96-well plates. Single-cell suspension of *Mtb* strains were added to a final volume in each well of 200 μL and an $OD_{580}$ of 0.01. Plates were kept at 37 °C and $OD_{580}$ readings were carried out after 14 days of incubation. Sauton's minimal medium supplementation with the corresponding solvents at the higher concentration constituted the control and was used to normalize the final $OD_{580}$ values.

**Immunoblot analysis of cytosolic proteins.** Protein extracts were prepared from WT, Δ*ndh-2*, and Δ*ndh-2*::*lbnox*-Flag cultures. Bacteria were harvested by centrifugation, washed with phosphate-buffered saline (PBS) with 0.05% Tween 80 and resuspended in 500 μL PBS, 1xprotease inhibitor cocktail (Roche). Bacterial lysis was performed by bead-beating three times at 4500 rpm for 30 s with 0.1 mm Zirconia/Silica beads. After centrifugation ($11,000 \times g$/10 min, 4 °C), the supernatant was filtered through a 0.2-μm SpinX column (Corning). Protein concentrations were determined using a DC Protein Assay Kit (Bio-Rad). For immunoblots, 30 μg protein were loaded and separated through SDS-PAGE, transferred to nitrocellulose membranes and incubated with anti-Flag (Sigma-Aldrich; 1:400 dilution) and anti-PrcB (1:15,000 dilution). Odyssey Infrared Imaging System (LI-COR Biosciences) was used to detect the proteins.

**NADH/NAD⁺ratio determination.** Strains were grown in modified Sauton's minimal medium until an $OD_{580}$ of 0.5, and resuspended in the same medium supplemented with DMSO or oleic acid (500 μM). Bacteria were harvested after 6 h of further incubation. The NADH/NAD + ratio was determined using the commercial kit Fluoro NAD (Cell Technnology), following the manufacturer's instructions. Protein quantification via Pierce BCA Protein Assay Kit (Thermo Scientific) was used to normalize NADH and NAD + concentrations.

**Drug susceptibility profiling.** Depending on the strain, bacteria were cultured in 7H9, modified Sauton's minimal medium, or modified Sauton's minimal medium supplemented with oleic acid (200 μM) until mid-log phase ($OD_{580}$ 0.6–0.8) in 100 mL roller bottle cultures. Bacteria were washed once in fresh medium and single-cell suspensions were prepared to a final $OD_{580}$ nm of 0.01 in the same medium. Compounds were solubilized in DMSO and dispensed into 384-well plates using a D300e Digital Dispenser (HP). DMSO at a final concentration of 1% was used as no-drug control. In all, 50 μL of single-cell suspension was pipetted to each well and cultures were incubated for 12 days at 37 °C. Final $OD_{580}$ values were normalized to no-drug control.

**Mouse infection.** All mouse experiments were performed in accordance with the Guide for the Care and Use of Laboratory Animals of the National Institutes of Health, with approval from the Institutional Animal Care and Use Committee of Weill Cornell Medicine. Female C57BL/6 mice (Jackson Labs) were infected with ~100–200 CFU/mouse using an Inhalation Exposure System (Glas-Col). Single-cell suspensions of early mid-log phase bacteria were prepared in PBS with 0.05% Tween 80, and then resuspended in PBS. When indicated, doxycycline containing food (2,000 p.p.m., Research Diets) was given to mice starting at the indicated time-points to induce protein depletion. Lungs and spleen were homogenized in PBS and plated on 7H10 plates or, when necessary, in solid modified Sauton's minimal medium to determine CFU/organ at the indicated time points.

**Marmoset Infection.** Common marmosets (Callithrix jacchus) breeding and all procedures were performed in accordance with the recommendations of the Guide for the Care and Use of Laboratory Animals of the National Institutes of Health. The NIAID Animal Care and Use Committee approved the experiments described herein (Protocol LCIM-9, Permit issued to NIH as A-4149-01). Prior to infection, marmosets were transferred to a BSL-3 animal facility approved for the containment of *Mtb* and handled as previously described[27,28]. Marmosets were aged between 2 and 5 years, of both genders, and ranged in weight from ~300 to 500 g at infection with *Mtb* strain CDC1551 with a nose-only aerosol generated in a BANG nebulizer through a CH Technologies inhalation system (Westwood, NJ). The dose was 10–25 CFU per animal. Infection progress was monitored by PET/CT, scanned twice prior to starting the treatment and every 2 weeks during 8 weeks of treatment. *Mtb* exposure in this experiment resulted in an average of 10 lesions/marmoset (SD = 4) as assessed by PET/CT at 5–6 weeks and 5 marmosets among 15 individuals were randomly assigned receive ND-10885. At necropsy, samples of all PET/CT identified lung lesions, pulmonary lymph nodes, and other possibly involved organs were weighed, homogenized, and plated in triplicate onto M7H11 agar supplemented with albumin, oleic acid, dextrose, cycloheximide (0.03 μg/mL), carbenicillin (0.05 μg/mL), polymixin B (25 μg/mL), trimethoprim (20 μg/mL), and activated charcoal powder (0.4% w/v) to enumerate bacterial CFU/structure.

**Plasma pharmacokinetics, dose finding, and drug treatment in marmosets.** The marmoset dose of ND-10885 was optimized in order to achieve a ratio of Area Under the Curve to MIC (AUC/MIC) well above the AUC/MIC at which maximum efficacy was observed in a murine model of *Mycobacterium avium* infection[29]. This corresponded to an AUC of 50 μg h/mL. In marmosets, ND-10885 appears to undergo CYP-mediated metabolism (Supplementary Fig. 9) and solubility limited bioavailability. To increase exposure while alleviating the need for higher dosing, we co-administered 1-aminobenzotriazole (ABT, Cayman Chemical, Ann Arbor, MI) at 20 mg/kg[31].

ND-10885 was formulated as an aqueous suspension stepwise in 10% w/v 2-hydroxypropyl-β-cyclodextrin, 10% w/v lecithin, 15% w/v saccharine, and 3% v/v Grape flavoring (Paddock Laboratories, Minneapolis, MN) and was administered at 1 mL/kg as an oral suspension for 2 months (50 mean dose days) to fed, hand-caught, awake, and restrained marmosets as previously described[28]. Doses used in pharmacokinetic and tolerability experiments included 20 mg/kg and 40 mg/kg with or without 20 mg/kg ABT administered both the previous day and 30 min prior to ND-10885. Discolored urine suggesting possible myoglobinuria was observed near the termination of the 20-dose tolerability study at 20 mg/kg, thus lower doses were used during the efficacy study. We employed a therapeutic drug monitoring approach during treatment to allow for dose escalation if the target AUC of 50 μg h/mL was not achieved. Treatment commenced with 20 mg/kg ABT 1 day prior and continued daily 30 min prior to the ND-10885 dose. Two marmosets began treatment with once daily 5 mg/kg ND-10885 and were escalated to 12.5 mg/kg daily (BI61 and M269). BI31 began daily 5 mg/kg ND-10885, was escalated to 12.5 mg/kg daily, and finally to 12.5 mg/kg twice daily in order to achieve an average AUC of 50 μg h/mL. Two marmosets were treated with the maximum dose of 12.5 mg/kg twice daily (BL24 and BI73). At the time of necropsy, tissues and lung lesions were sampled and processed for quantitation of ND-10885 as previously described[27,28,53].

**PET/CT data analysis.** PET/CT files were co-registered and serial scans were aligned in MIM Maestro (v. 6.2, MIM Software Inc, Cleveland, Ohio), and each lesion was individually defined within a three-dimensional region of interest (ROI) on each serial scan. The numerical output of these ROIs including the radiodense volumes (−100 to 200 HU) from the CT images and [18F]-2-fluoro-2-deoxyglucose (FDG) uptake (expressed as TGA) for the ROI voxels with a standardized uptake value (SUV) ≥ 2 were assessed for change in subsequent scans[28]. Three readers performed the extractions independently for each set of animal lesions and the data presented are an average of the resulting readings. The CFU data for each lesion or structure were aligned to the PET/CT and histology data to using the prepared necropsy maps and logs to evaluate the response of the individual lesion types during treatment.

**Quantitation of ND-10885 in plasma and TB lesions.** ND-10885 levels in plasma, tissues, and lung lesions were measured by LC-MS/MS in electrospray positive-ionization mode (ESI+) on a Sciex Qtrap 4000 triple-quadrupole mass spectrometer combined with an Agilent 1260 HPLC using Analyst software. Chromatography was performed with an Agilent Zorbax SB-C8 column (2.1 × 30 mm; 3.5-μm particle size) using a reverse phase gradient elution. In all, 0.1% formic acid in Milli-Q deionized water was used for the aqueous mobile phase and 0.1% formic acid in acetonitrile (ACN) for the organic mobile phase. Multiple-reaction monitoring (MRM) of parent/daughter transitions in electrospray positive-ionization mode (ESI+) was used to quantify ND-10885. DMSO stock of ND-10885 was serial diluted in blank K2EDTA plasma (Bioreclammation) to create standard curves and quality control samples. ND-10885 was extracted by combining 20 μL of spiked plasma or study samples and 200 μL of acetonitrile/methanol 50/50 protein precipitation solvent containing 20 ng/mL Verapamil internal standard (IS). Extracts were vortexed for 5 min and centrifuged at 4000 RPM for 5 min. The supernatants were analyzed by LC-MS. Verapamil IS was sourced from Sigma-Aldrich. The following MRM transitions were used for ND-10885 (322.1/133) and Verapamil (455.4/165.2). Sample analysis was accepted if the concentrations of the quality control samples were within 20% of the nominal concentration.

**Quantification and statistical analysis.** Generation of graphics and data analyses were performed in Prism version 7.0 software (GraphPad), Spotfire version 7.8.0 software (TIBCO), and MIM Encore (Version 6.7.8, MIM Software Inc.).

**Reporting summary.** Further information on research design is available in the Nature Research Reporting Summary linked to this article.

## Data availability
The authors confirm that all relevant data are included in the paper and/or its supplement. Sequencing data can be accessed in NCBI's Sequence Read Archive (SRA) database under the BioProject ID PRJNA532433. Plasmids and strains generated in this study will be available through material transfer agreements upon reasonable request to the corresponding author.

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

## Acknowledgements

We thank Natalia Betancourt, Shuang Song, and Toshiko Odaira for help with mutant construction, Claire Healy for help with mutant characterization, Becky Sloan, Daniel Schimel, and Ashley Majewski for their technical expertize, Garrett C. Moraski and Marvin J. Miller for providing ND-10885, and Richard Herbert DVM for supervision of the NIAID marmoset experiments. We acknowledge James M. Bean from MSKCC and the use of the Integrated Genomics Operation Core at MSKCC, funded by the NCI Cancer Center Support Grant (CCSG, P30 CA08748), Cycle for Survival and the Marie-Josée and Henry R. Kravis Center for Molecular Oncology. This work was supported by the Tri-Institutional TB Research Unit (NIH grant U19AI111143), the Bill & Melinda Gates Foundation's TB Drug Accelerator Program (grants OPP1024065 (D.S.), OPP1024021 (C.E.B. through the Foundation of the NIH), OPP1174780 (V.D.)), and the intramural research program of the NIAID, NIH (C.E.B.). T.B. was supported by a Potts Memorial Foundation fellowship.

## Author contributions

T.B., K.O.B., D.T., C.A.E., S.W., L.E.V., D.M.W., M.L.S., E.K.D., M.Z., and B.P. performed experiments. T.M. helped to provide ND-10885. P.V.D., and T.M. analyzed metabolite data for ND-10885. J.A., H.J.Y., L.E.V., and C.E.B. analyzed the radiology and pathology data. T.B., L.E.V., H.I.B., V.D., S.E., C.E.B., and D.S. wrote the manuscript.

## Competing interests

The authors declare the following competing interests: TM and PVD are employees of Eli Lilly and Company.
