## [Peer Review File · Nature Communications]

Reviewers' comments:

Reviewer #1 (Remarks to the Author):

The paper by Schnappinger and colleagues is an essential and outstanding piece of work for researchers in the field of TB drug discovery.

It is the first paper to comprehensively genetically dissect the role of the respiratory complexes cytochrome bc1-aa3/cytochrome bd oxidases and the non-proton translocating type II NADH dehydrogenase (NDH-2). All essentiality (transposon based) screens in *M. tuberculosis* and various gene deletion studies to date have suggested that cytochrome bc1-aa3 and NDH-2 are essential for growth, but the data have never really been that convincing or satisfying – too many problems with the gene deletions and not being able to generate the mutant being used to imply essentiality. Finally (we waited a long time), the conditional gene silencing DUC system generated in the Schnappinger laboratory has been used to convincingly nail these long-standing questions and importantly provide new opportunities and insight for TB drug development.

Specific points to address:

1. Line 60: It is often not appreciated that some of these drugs activate (not inhibit) NDH-2 activity leading to rapid cell death. I feel the jury is still out on clofazimine. This should be mentioned. Please cite: Heikal, A., Hards, K., Cheung, C.-Y., Menorca, A., Timmer, M.S., Stocker, B.L. and Cook, G.M. Activation of type II NADH dehydrogenase by quinolinequinones mediates antitubercular cell death. *Journal of Antimicrobial Chemotherapy* 71:2840-2847 (2016) PMID: 27365187.
2. It has also been shown that manipulating the ratio of menaquinone/menaquinol via reducing agents can accelerate cell death by conventional TB drugs: Vilcheze C, Hartman T, Weinrick B, Jain P, Weisbrod TR, Leung LW, Freundlich JS, Jacobs WR, Jr. 2017. Enhanced respiration prevents drug tolerance and drug resistance in *Mycobacterium tuberculosis*. *Proc Natl Acad Sci U S A* 114:4495-4500.
3. The first major finding of the paper is that the cytochrome bc1-aa3 is not essential for growth, yet inhibitors (e.g. Q203) of this complex still inhibit growth at nM concentrations – how do the authors explain this? The authors use 7H9 medium in their knockdown experiments – did the authors investigate the essentiality of cytochrome bc1-aa3 as a function of carbon source?
4. On the basis of cytochrome bc1-aa3 not being essential for growth in *M. tuberculosis*, is cytochrome bd upregulated in these knockdowns and deletion mutants? Some quick qPCR experiments would add value here – mechanistically this is an important result for the paper. The same question applies – why don't cells upregulate cytochrome bd when challenged with Q203? Data suggests Q203 must also inhibit cytochrome bd if the former experiment shows upregulation of *cydAB*. Why doesn't Q203 inhibit the cytochrome bc1-aa3 deletion mutants (Figure 6g)? Does this mean that when you shut down cytochrome bc1-aa3 with Q203 the cells can't upregulate *CydAB*.....?
5. Line 107: I feel this statement needs a little more context. It has been shown that inhibiting both oxidases is required to clear *Mtb* infection in mice (as shown in reference 18). It is probably worthy of noting and citing that Q203-like molecules show synergy in animal models with RIF and PZA – cite *ACS Infect Dis.* 2019 Feb 8;5(2):239-249. doi: 10.1021/acsinfecdis.8b00225. Epub 2018 Dec 11. (PMID:30485737, DOI: 10.1021/acsinfecdis.8b00225). Did the authors try any synergy experiments in mice with ND-10885?
6. Can the authors comment on the MIC of ND-10885 versus Q203 – there is a large difference in potency. Do the authors have an explanation for the increased cavitory lung lesions with ND-10885 treatment? Were any bacteria recovered to determine the development of resistance to ND-10885? Does resistance develop to ND-10885 in vitro?

7. The new data with *ndh2* deletions is fascinating especially the observation that NDH-2 is obligatory for growth with fatty acids in the medium. So, one might argue it would be a great drug target for *Mtb* in vivo as it will likely encounter a fatty acid-rich environment. Although the data in Figure 4 suggests otherwise. Is the difference between the *ndh* mutant and wt in Figure 4C (hypoxia) significant?

8. What would have been good in these experiments is to determine the effect of the triple deletion i.e. *ndh* and *nuo* all deleted? Can the authors comment? Furthermore, is the *nuo* operon upregulated in the *ndh* mutant? This is important information from a mechanism point of view. A key experiment not discussed here would be to determine if *cydAB* is upregulated in the *ndh* mutant. The rationale here is that using two proton pumping machines (*Nuo* and *bc1-aa3*) might be antagonistic. One might predict that *bd* is upregulated in the *ndh* mutant to offset the pmf. ATP synthase may also be upregulated for the same reason to consume excess pmf.

9. The authors identify a number of fatty acids toxic to *Mtb* in the absence of *ndh2* – the authors propose a mechanistic model centered around altered NADH/NAD⁺ ratios – agree this is not the mechanism. What is the mechanism of oleic acid inhibition of growth in Figure 5C – I am assuming this is NOT a pH effect? Figure 5g is addressing my point above about excess pmf but I am not sure CCCP is a good tool to do this. I would try either valinomycin or nigericin as that would pinpoint whether it is the electrical potential or delta pH component of the pmf respectively – CCCP does not discriminate.

10. I would argue that upregulation of non-proton translocating cytochrome *bd* would escape the back pressure of Complex I. Do the authors have that expression data for *cydAB* in the delta *ndh* background? I would also propose that *bc1-aa3* would be downregulated in the delta *ndh* background. One might also hypothesize that you could achieve the deletion of *ndh2* in a delta *bc1-aa3* background in the PRESENCE of fatty acids.

11. Complex I inhibitor experiments Figure S8: the authors demonstrate synergy between the *bc1-aa3* knockout and rotenone. Does that argue that a double mutant *bc1-aa3/nuo* would not be viable and if so why would that be? The same questions apply to a *bd/ndh2* double deletion.

12. Figure 6: if indeed complex I is upregulated in the delta *ndh2* mutant (see comment above) it is not surprising that CFZ is still inhibitory as it will still compete for the same electrons off NADH (via Complex I) with menaquinone. CFZ inhibits growth through a redox cycling mechanism not by inactivating NDH-2 (see <https://www.ncbi.nlm.nih.gov/pubmed/21193400>).

Reviewer #2 (Remarks to the Author):

This manuscript addresses important questions regarding the relative roles of some partnering complexes or enzymes of the respiratory chain of *M. tuberculosis*.

This topic is of main importance in the context of the development of recent drugs targeting this machinery, with some in clinical development.

This work confirms a series of results reported previously on synthetic lethality and produce extra novel information enlightening our understanding of important inhibitors.

Whereas, globally well written, the document needs important improvements and clarifications:

Based on their observation that bc1-aa3 mutant is only partially attenuated in mice, authors suggest that inhibition of bc1 by a drug may not improve human chemotherapy:

First, the observation that bc1 inhibitors are only bacteriostatic and poorly effective in mice has been largely documented already. Here, authors wanted to demonstrate their strong and important assumption using the marmoset model. However, it is important to be cautious with the obtained data, as the treatment of marmoset by ND-10885 may not be fully comparable to the treatment with mechanistically related compounds such as Q203 (or arylvinylpiperazine amides).

Authors have chosen to switch from Q203 to ND-10885 because of PK and exposure problems. The choice of replacing Q203 by this compound has some possible consequences:

-) As shown on Sup fig 10, ND-10885 is far from being an ideal compound in term of stability in vivo. ND-10885 is shown prone to metabolic transformation by liver enzymes. Authors should add in the text (line 118-120) a link to the supplementary fig. 10 attracting attention on ND-10885 metabolism.

-) Part of the argumentation sits on the assumption that ND-10885 and Q203 share the same mode of action. However, it is very surprising that ND10885 has not been tested, with the long list of other compounds shown in table S1 and S2, against the various ETC mutants. It is essential to prove that ND10885 shows the same profile than Q203 against the various mutants, in particular an absence of effect against the Δ ctaE-qcrCAB mutant.

-) comparing the activity of Q203 to the one of ND-10885 on WT bacteria is also needed

-) The conclusions regarding the efficacy of ND10885 in animals must be exclusively restricted to this compound. Extrapolating these observations to other compounds targeting the same pathway should be cautiously discussed.

-) As this is the first time that a group relay a double KO Ndh-2 (Ndh+NdhA), which was suggested impossible by others (Vilchèze, PNAS2018), it would be important to confirm the genotype by WGS to complete Southern blot data.

In the same way, southern blots reporting on the deletion of bc1-aa3 are showing high background signal, which makes difficult to be formal on the deletions. In addition, the EcoR1-EcoR1 band observed in the Southern blot of the Δ ctaD clone is bigger than 1.5 kb, which do not match with the expected 1360 bp. As this part of the work is central in the manuscript, I would highly recommend to verify all mutants by WGS.

Line 86. There is 1.5 log difference between WT and Δ ctaC at day 110. I don't think one can write "similar titers" in this situation. Nonetheless, at the view of the progression of the curve of the Δ ctaC mutant, it is difficult to predict whether equivalent titers would be reached few weeks later. In addition, in Fig1, Δ ctaC-comp shows growth equivalent to the one of Δ ctaC. Thus, it does not seem correct to say that "The final CFU titers were lower for Δ ctaE-qcrCAB and Δ ctaC than for WT Mtb, which suggest that the cytochrome bc1-aa3 oxidase is indeed required for full virulence of M. tuberculosis". In addition, this sentence is contradictory with the previous one, claiming : "SIMILAR titers" between WT and mutant Δ ctaC.

It would be good to propose some hypothesis to explain the differences of "toxicity" of long lipids versus cholesterol for the NDH-2 mutant.

Some claims of novelty should be reformulated. For example "We conclude that Mtb requires the cytochrome bc1-aa3 supercomplex to achieve optimal growth rates and maximal titers, but that this complex is not essential for growth or persistence as long as Mtb can express a functional cytochrome bd oxidase". This fact has been previously reported by many others: for example, Kalia et al, or Foo et al, showed that the inhibition of bc1:aa3 by Q203 is not able to fully block the electron flow through the cytosol due to the presence of the cytochrome oxidase bd. Same authors reported "multiple lesions and inflamed foci were found in the lungs of the mice (...)

treated by Q203".

This raises the question of the novelty of some important data presented as novel in this manuscript such as "These findings raise concerns about the use of cytochrome bc1-aa3 oxidase inhibitors to treat TB."

Authors show that NDH-2 mutants are only slightly attenuated in mice (fig 4D-E). Due to the very different type of growth, metabolisms and lesions, and thus carbon source in mice versus marmosets, and the relationship shown here between lethality of Δ NDH-2 and lipids, it would be much more relevant to test NDH-2 mutants in marmosets.

Minor points:

The chemical structure of ND-10885 should be shown in the main text, also with Q203, to show that these two compounds have substantial dissimilarities.

Fig.S1. Put the mean of the experiments instead of "one representative of at least two".

Fig2C and D could be transferred in supplementary. Their reading is quite challenging and it is almost impossible for the reader to make its own general conclusion out of it.

Fig5D: not needed

Fig5E can be transferred in supplementary data

Line 215: structures of DDD00853663, DDD00946831 should be included in sup. data

FigS6. The probe is missing in part B (WT)

Line 120: add sup fig. 10

Not enough data (supplementary) are given regarding genetic constructions leading to gene deletions. Experiments should be sufficiently detailed to be reproducible by colleagues.

Reviewer #3 (Remarks to the Author):

In the current study the authors have tried to address the non-redundancy of Mtb respiratory chain components and tried to elucidate the associated non-essentiality/dispensability of these components for the survival of Mtb. The significance of the studies conducted here is very high and the paper is well written. Also, considering that drug Q203 is in phase 2 clinical trial, this data serves best fit pre-clinical data integrating molecular biology, mouse and nonhuman primate studies. In general I felt that there could have been a better discussion of future directions at the end of the manuscript. Overall a manuscript that after revisions will be highly improved and worthy.

While the authors have done commendable amount of work, the manuscript however has some issues and shortcomings which I will try to point out:

Major issues:

1. Line 47: While the authors briefly mention a previous study, which reported M.tb lacking both D-ndh/ndhA to be not viable, they do not address the finding in a convincing way. Neither are the associated results and molecular methodology explained in enough details to support their contrasting claim, while the cited study does describe their methods and approaches in details enough to make the reader confident in the reported results.
2. Line 58: While the study on clofazimine does interrogate activation by NDH-2, it doesn't rule out the non-redundancy of this reaction by other oxidoreductases which needs to be mentioned in this context.
3. The animal experiment lacks mention of statistical method involved and the extent of significance, and is thus not rigorous.
4. Fig 2 & 3: All marmoset data lacks infected controls. Additionally, Figure 2 mentions Klebsiella superinfection? Furthermore, Fig 2A shows significant difference in only 2 lesions and Fig 2B in

only 3. The inferences drawn are non-convincing and not strong.

5. Fig 3A: Historical data for different regimens appear to have completely different dynamics and therefore makes the comparison inappropriate. Also, curiously 2-drug streptomycin-isoniazid (HS) regimen outperforms the 4-drug isoniazid-rifampicin-pyrazinamide-ethambutol (HRZE) regimen.

6. In comparing the images from the current manuscript with a previous paper from some of these authors published in collaboration with Dr. Joanne Flynn, it appears that previously distinct mediastinal lymph node involvement was observed in Mtb infected animals, whereas those receiving Q203 don't appear to have any. Although Mtb CFUs delivered were listed in the previous paper and are not described in the current study so it could be a function of that but if not then this drug could be a game changer for treatment of EPTB, and perhaps the authors could discuss that? The authors recognize that that bd oxidase is dependent on the oxygen tension and is strongly upregulated during hypoxia. The PET CT scan mentions only about lung lesions where the oxygen tension is high. Analysis of lesion characteristics in relatively anaerobic tissue like mediastinal or cervical lymphnodes will further clarify efficacy of this drug in extrapulmonary TB conditions which are relatively difficult to manage with currently prescribed drug regimens.

7. Though the authors intend to compare the current study with their previous study involving 4 drug chemotherapy, starting the therapy 6 weeks after infection suggests that it was too late and too less in this case. A previous study by (Cadena et.al.) suggest that even with low dose infection with 7 CFU CDC1551 cavitory lesions were reported at 4 weeks, initiating the therapy early could have seen better results.

8. Line 110 : "Treatment of Mtb-infected marmosets with ND-10885 controlled infection and inflammation, but increased the occurrence of cavitory lesions" is misleading. As shown in fig 3. No. of cavitory lesions and caseous necrosis has only increased in the ND-10885 group. It is insufficient to conclude on infection and inflammation status only on basis PET-CT findings of lung lesion glycolysis and hard volume, adding biochemical data like CRP or other inflammatory markers would help corroborate the inflammation status better.

9. Survival in control group of previous study (Via LE) was around 9-10 weeks whereas according to Fig 2.a.) and 2b.) survival seems to be 111 days. This shows that the treatment drug clearly has superior survival benefits than control group. A survival analysis curve involving all the animals in fig 3. will give a better representation in this context.

Minor issues:

1. The methods are not well described and need substantial improvement to make the results appear convincing. The molecular strategy needs to be elaborated in detail and include accurate mentions of plasmids, primers and validations of successful interventions. Furthermore, the should also expand on the methodology employed for estimating CFU and growth rates throughout the study. Sticking to a general presenting style will make the data appear more understandable(preferably log scale which is customary of the field for in vivo animal pathogenic burden and CFU/ML for in vitro).

2. The introduction is disparate and not continuous. The published research in the field need to be mentioned and cited in context of current rationale. Most of the citations need to be revised for accuracy.

3. Line 68-70: Citations and prior reports suggesting NDH-1 and its probable compensation for NDH-2 needs to be mentioned and discussed.

4. Results: Fig 1A & B: Why does the complemented strains appear to outperform even wild type in murine model? Were the strains validated for confirming they are still not merodiploid? Were the cloned enzymes being overexpressed? If not, why does the complemented strain has a survival advantage? Also, why the same strain has a slightly diminished growth than WT in vitro Suppl. Fig 1D??

5. Fig 1C&D: It is surprising to see that the Δ ctaC strain is diminished for growth. This should be discussed. If there is any data on the in vivo burdens of Δ ctaD-qcrCAB or Δ ctaC-qcrCAB or Δ qcrCAB, the authors are encouraged to show that. Alternatively this data could be moved to the supplementary material.

6. Suppl Fig 4: This figure depicts one of the most convincing findings of this study and clearly

establishes the compensatory roles of cytochrome bd oxidase and cytochrome bc1-aa3 complex for in vivo survival in mice model. Statistics needs to be added though.

7. Line 123: What was estimated infection dose in CFU for the low dose aerosol infection in marmosets? This detail should be provided.

8. Fig 3C: It would be best if quantitative analysis of multiple fields was performed.

9. Line 368-371: Whether the complementation was done under the control of self-promoter or any constitutively expressing strong (hsp60/myc) promoter?

10. There is no information about the dose used for infection. This is an important gap that should be filled.

11. Fig3: no control data with infected untreated (3A). The "HS" seems to work better than HRZE treatment. Can the authors present any suitable explanation. Fig3B should be supplemented by an infected untreated control.

12. Line 117-118: is fundamental information about the activity of the drug ND-10885 against Mtb? E.g., MIC?

13. Line 47: A reference could be added on the study that showed that ndh was dispensable for growth of Mtb.

14. In Supplementary Figure 1 A- H, only panels D and H have their scales different on the Y axis, it could be aesthetically better to have all the scales same.

15. The gel picture in Panel D in Supplementary figure 2 is of a very low quality.

16. Why weren't the mutants (TetOFF and deletion mutants) not studied for growth for the same number of days? TetOFF were monitored for growth up to 20 days while Δ ctaE-qcrCAB were studied for 25 days and Δ cydABDC ctaE-qcrCAB-TetOFF was studied for more than 30 days. This should at the very least be described.

17. Line 87 seems a little vague with the authors concluding that the lower CFU titres are responsible for the lower virulence of Mtb. The authors have not shown CFU data (all figures are presented in OD) and no experiments were conducted at this point to correlate the CFU with any virulence parameter to enable this conclusion.

18. Lines 91 to 94 appear a little unnecessary in the results section, perhaps they should be moved to the discussion section.

19. To be consistent in the supplementary figures involving mutant creation, the authors should either show pictures of agar plates in all (Supp Fig S2., S3.) or not show them at all.

20. Supplementary Figure S4. The scales are not consistent on the panels (3 out of 4 have 125 days while A has 80 days on the X axis and two have 10^6 while 2 have 10^7 on the Y axis).

21. Line 116: What could be the reason for the Q203 not being able to provide acceptable exposure in marmosets?

22. Figure 3C: The pathology panel has some factors that needs improvement. The first picture on top left has a small box that needs to be connected with some arrows or lines to the bottom enlarged picture to easily demonstrate that the bottom picture is an enlargement of that small box. As it stands now, unless someone reads the legend, will think these are different pictures.

23. The contrast of the figure on the bottom needs to be improved as right now it is difficult to visualize the marked regions clearly.

We thank the reviewers for carefully reading our manuscript and for the constructive criticisms. We have addressed the issues raised by the reviewers point by point and believe the manuscript has been greatly improved. Please find below the reviewers' comments followed by our responses.

Reviewer #1

The paper by Schnappinger and colleagues is an essential and outstanding piece of work for researchers in the field of TB drug discovery. It is the first paper to comprehensively genetically dissect the role of the respiratory complexes cytochrome bc1-aa3/cytochrome bd oxidases and the non-proton translocating type II NADH dehydrogenase (NDH-2). All essentiality (transposon based) screens in *M. tuberculosis* and various gene deletion studies to date have suggested that cytochrome bc1-aa3 and NDH-2 are essential for growth, but the data have never really been that convincing or satisfying – too many problems with the gene deletions and not being able to generate the mutant being used to imply essentiality. Finally (we waited a long time), the conditional gene silencing DUC system generated in the Schnappinger laboratory has been used to convincingly nail these long-standing questions and importantly provide new opportunities and insight for TB drug development.

We very much thank the reviewer for his/her enthusiasm and appreciation of our work.

Specific points to address:

1. Line 60: It is often not appreciated that some of these drugs activate (not inhibit) NDH-2 activity leading to rapid cell death. I feel the jury is still out on clofazimine. This should be mentioned. Please cite: Heikal, A., Hards, K., Cheung, C.-Y., Menorca, A., Timmer, M.S., Stocker, B.L. and Cook, G.M. Activation of type II NADH dehydrogenase by quinolinequinones mediates antitubercular cell death. *Journal of Antimicrobial Chemotherapy* 71:2840-2847 (2016) PMID: 27365187.

2. It has also been shown that manipulating the ratio of menaquinone/menaquinol via reducing agents can accelerate cell death by conventional TB drugs: Vilcheze C, Hartman T, Weinrick B, Jain P, Weisbrod TR, Leung LW, Freundlich JS, Jacobs WR, Jr. 2017. Enhanced respiration prevents drug tolerance and drug resistance in *Mycobacterium tuberculosis*. *Proc Natl Acad Sci U S A* 114:4495-4500.

We agree that the concepts stated in points 1 and 2 are important for a more complete understanding on how CFZ might impact *Mtb*. We have edited the manuscript and added the suggested references (lines 57-59).

3. The first major finding of the paper is that the cytochrome bc1-aa3 is not essential for growth, yet inhibitors (e.g. Q203) of this complex still inhibit growth at nM concentrations – how do the authors explain this? The authors use 7H9 medium in their knockdown experiments – did the authors investigate the essentiality of cytochrome bc1-aa3 as a function of carbon source?

We thank the reviewer for raising this point. In our experience, Q203 does not completely inhibit *Mtb*'s growth. As it can be observed in the figure below, although it grows more slowly, wild-

type *Mtb* can still grow at the higher Q203 concentrations tested. The fact that a plateau is achieved at the high end of the concentrations range makes it unlikely that increasing Q203 concentration will result in a truly bacteriostatic effect.

We did not evaluate the conditional essentiality of cytochrome *bc1-aa3* oxidase as a function of carbon source, but we intend to address this issue in a future study.

4. On the basis of cytochrome *bc1-aa3* not being essential for growth in *M. tuberculosis*, is cytochrome *bd* upregulated in these knockdowns and deletion mutants? Some quick qPCR experiments would add value here – mechanistically this is an important result for the paper. The same question applies – why don't cells upregulate cytochrome *bd* when challenged with Q203?

The reviewer correctly points out a mechanistic explanation for the non-essentiality of cytochrome *bc1-aa3* oxidase. In response to the reviewer's suggestion we have measured transcript levels of *cydA* (cytochrome *bd* oxidase) in mid-exponential cultures, as well as of other respiratory chain related genes: *nuoA* (NDH-1), *ndh* and *ndhA* (NDH-2), *ctaC* (cytochrome *aa3* oxidase), *qcrA* (cytochrome *bc1* reductase) and *atpE* (ATP synthase). We have added the qPCR profiles as Supplementary Fig. 4 (see lines 94-101). Our results show that both $\Delta ctaC$ and $\Delta ctaE$ -*qcrCAB* overexpress *cydA*, confirming the hypothesis proposed by the reviewer.

This is consistent with a previous study that showed that challenging *Mtb* with Q203 leads to an overexpression of cytochrome *bd* oxidase encoding genes [Arora, K., et al. (2014). "Respiratory flexibility in response to inhibition of cytochrome C oxidase in *Mycobacterium tuberculosis*." *Antimicrob Agents Chemother* 58(11): 6962-6965]. This citation has been added to the manuscript (line 99).

In addition, the qPCR profiles showed an up-regulation of *ndh* in both cytochrome *bc1-aa3* deletion mutants. We do not have a clear explanation for this *ndh* expression profile, but one can speculate that the overexpression of NDH-2 might increase the influx of electrons to the respiratory chain, compensating for the less efficient alternative terminal cytochrome *bd* oxidase.

Data suggests Q203 must also inhibit cytochrome *bd* if the former experiment shows upregulation of *cydAB*. Why doesn't Q203 inhibit the cytochrome *bc1-aa3* deletion mutants

(Figure 6g)? Does this mean that when you shut down cytochrome *bc1-aa3* with Q203 the cells can't upregulate *CydAB*?

As pointed out by the reviewer, cytochrome *bc1-aa3* oxidase deletion mutants are not susceptible to Q203. This suggests that Q203 does not have other relevant targets *in vitro* and rules out a possible effect on cytochrome *bd* oxidase, since terminal oxidases are synthetically lethal. Again, we think that the key point here is that Q203 is not truly bacteriostatic, as we have previously discussed in point 3; instead, Q203 just slows down growth, since only cytochrome *bd* oxidase is operating in the respiratory chain. It seems also worth noting, that induction of cytochrome *bd* oxidase in response to deletion of cytochrome *bc1-aa3* oxidase will contribute to lack of efficacy of Q203. In well-aerated WT *Mtb*, induction of cytochrome *bd* oxidase will only occur in response to inhibition of cytochrome *bc1-aa3* oxidase leading to decreased efficacy over time.

5. Line 107: I feel this statement needs a little more context. It has been shown that inhibiting both oxidases is required to clear *Mtb* infection in mice (as shown in reference 18). It is probably worthy of noting and citing that Q203-like molecules show synergy in animal models with RIF and PZA – cite ACS Infect Dis. 2019 Feb 8;5(2):239-249. doi: 10.1021/acsinfecdis.8b00225. Epub 2018 Dec 11.(PMID:30485737, DOI: 10.1021/acsinfecdis.8b00225). Did the authors try any synergy experiments in mice with ND-10885?

We have not conducted synergy experiments in mice, but it is certainly a relevant idea in the context of possible improvements to current drug regimens. We have edited the manuscript [see lines 116-118] and added the suggested reference.

6. Can the authors comment on the MIC of ND-10885 versus Q203 – there is a large difference in potency. Do the authors have an explanation for the increased cavitory lung lesions with ND-10885 treatment?

Q203 is 10-fold more potent than ND-10885 against CDC1551. The results are consistent with a continued growth of organisms in caseous hypoxic lesions where the organism is not using *QcrB*. Increasing bacterial numbers in the caseous center appears to allow liquification of the caseum and cavity formation.

Were any bacteria recovered to determine the development of resistance to ND-10885? Does resistance develop to ND-10885 *in vitro*?

In response to the reviewer questions we analyzed 72 colonies obtained from 8 different lesions. None were resistant to ND-10885. This data was added in Supplementary Table 2.

7. The new data with *ndh2* deletions is fascinating especially the observation that NDH-2 is obligatory for growth with fatty acids in the medium. So, one might argue it would be a great drug target for *Mtb* *in vivo* as it will likely encounter a fatty acid-rich environment. Although the data in Figure 4 suggests otherwise. Is the difference between the *ndh* mutant and wt in Figure 4C (hypoxia) significant?

We were also surprised by the fact that the $\Delta ndh-2$ fatty acid sensitivity did not translate into an *in vivo* essential phenotype. It is possible that the fatty acid concentrations encountered by *Mtb*

in vivo are not high enough to reach the toxicity threshold. Also, the fact that $\Delta ndh-2$ is less sensitive to cholesterol may have also contributed to the ability to grow and survive in mice. Nevertheless, $\Delta ndh-2$ growth was mildly attenuated, which is indicative of some toxicity, but not enough to completely block growth.

Regarding Fig. 4C, we did a statistical test and the differences are significant. Fig. 4C and corresponding legend were updated.

8. What would have been good in these experiments is to determine the effect of the triple deletion i.e. *ndh* and *nuo* all deleted? Can the authors comment?

That the NDH-1 inhibitor rotenone is bactericidal for $\Delta ndh-2$, yet has no effect on wt Mtb, strongly suggested that NDH-1 and NDH-2 are synthetically lethal. This means that the triple mutant would most likely be not viable.

Furthermore, is the *nuo* operon upregulated in the *ndh* mutant? This is important information from a mechanism point of view. A key experiment not discussed here would be to determine if *cydAB* is upregulated in the *ndh* mutant. The rationale here is that using two proton pumping machines (Nuo and *bc1-aa3*) might be antagonistic. One might predict that *bd* is upregulated in the *ndh* mutant to offset the pmf. ATP synthase may also be upregulated for the same reason to consume excess pmf.

We agree with the hypothesis raised by the reviewer. To address this, we have measured the transcript levels of several respiratory chain related genes (*ndh*, *ndhA*, *nuo*, *cydA*, *ctaC*, *qcrA*, and *atpE*) in mid-exponential cultures. The results were added as Supplementary Fig. 4 (see lines 182-186). Contrarily to the prediction, *nuoA*, *cydA* and *atpE* were not differentially expressed in $\Delta ndh-2$. The remaining genes also did not show a differential expression in the mutant. However, we agree that a balance between proton pumping and non-proton pumping enzymes is likely to occur to avoid a disturbance on pmf. One can speculate that this might be achieved, for example, through allosteric modulation of respiratory chain enzymes activity.

9. The authors identify a number of fatty acids toxic to Mtb in the absence of *ndh2* – the authors propose a mechanistic model centered around altered NADH/NAD⁺ ratios – agree this is not the mechanism. What is the mechanism of oleic acid inhibition of growth in Figure 5C – I am assuming this is NOT a pH effect?

Figure 5g is addressing my point above about excess pmf but I am not sure CCCP is a good tool to do this. I would try either valinomycin or nigericin as that would pinpoint whether it is the electrical potential or delta pH component of the pmf respectively – CCCP does not discriminate.

We very much thank the reviewer for this comment, as indeed CCCP does not discriminate between the pmf components. Hence, we have followed the reviewer's suggestion, and performed the same experiment with valinomycin and nigericin, as well as rifampicin as a control. We have substituted the CCCP data for these new profiles, as we feel these experiments are much more informative (see Fig. 5 E, F G, lines 208-213, and lines 284-288). As it can be observed only valinomycin could rescue the phenotype, pointing out to a de-regulation of membrane potential.

Putting the data together, we propose the following mechanism. β -oxidation of highly reduced carbon sources, such as oleic acid, leads to the generation of NADH. The finding that the NADH

oxidase LbNox is capable of fully rescuing oleic acid sensitivity clearly shows that NADH plays a role in this phenotype. Since NDH-1 is the only functional NADH dehydrogenase, NADH re-oxidation is always coupled with proton pumping, which may lead to a de-regulation in either the membrane potential or ΔpH . The rescue of the phenotype with valinomycin, but not with nigericin, shows that a de-regulation of the membrane potential is, at least in part, responsible for oleic acid toxicity.

10. I would argue that upregulation of non-proton translocating cytochrome bd would escape the back pressure of Complex I. Do the authors have that expression data for *cydAB* in the delta *ndh* background? I would also propose that *bc1-aa3* would be downregulated in the delta *ndh* background.

We agree that cytochrome bd up-regulation and cytochrome *bc1-aa3* down-regulation would provide a good explanation for the adaptation of the respiratory chain to the lack of a functional NDH-2. We tested this experimentally, but were unable to detect changes in *cydA*, *ctaC* or *qcrA* transcripts (Supplementary Fig. 4; see lines 182-186). We cannot rule out that activity of the oxygen dependent oxidase occurs by post-transcriptional mechanisms.

One might also hypothesize that you could achieve the deletion of *ndh2* in a delta *bc1-aa3* background in the PRESENCE of fatty acids.

We agree with the reviewer. To get some insight on this matter we have checked if Q203 was able to restore growth in the presence of oleic acid. The Figure below represents the growth of $\Delta\text{ndh-2}$ challenged with a range of Q203 concentrations in different media: fatty acid free medium (control) and media supplemented with different concentrations of oleic acid. As it can be observed, Q203 was not able to make medium supplemented with oleic acid permissive to $\Delta\text{ndh-2}$ growth, which suggests that an NDH-2/cytochrome *bc1-aa3* deleted mutant is unlikely to grow in the presence of fatty acids.

11. Complex I inhibitor experiments Figure S8: the authors demonstrate synergy between the *bc1-aa3* knockout and rotenone. Does that argue that a double mutant *bc1-aa3/nuo* would

not be viable and if so why would that be? The same questions apply to a *bd/ndh2* double deletion.

Our chemical-genetic interaction data, namely susceptibility of Δ *ctaE-qcrAB* to rotenone (NDH-1 inhibitor) and Δ *cydABDC*/ 2-mercaptoquinazolinones (NDH-2 inhibitor), are consistent with the hypothesis proposed by the reviewer.

12. Figure 6: if indeed complex I is upregulated in the delta *ndh2* mutant (see comment above) it is not surprising that CFZ is still inhibitory as it will still compete for the same electrons off NADH (via Complex I) with menaquinone. CFZ inhibits growth through a redox cycling mechanism not by inactivating NDH-2 (see <https://www.ncbi.nlm.nih.gov/pubmed/21193400>).

We agree. This can explain why CFZ is still active against Δ *ndh-2*, although the paper referenced by the reviewer showed that other microorganisms that predominantly express NDH-1 do not reduce CFZ (lines 298-300). This of course does not rule out that *Mtb*'s NDH-1 can reduce CFZ.

Reviewer #2

This manuscript addresses important questions regarding the relative roles of some partnering complexes or enzymes of the respiratory chain of *M. tuberculosis*. This topic is of main importance in the context of the development of recent drugs targeting this machinery, with some in clinical development.

This work confirms a series of results reported previously on synthetic lethality and produce extra novel information enlightening our understanding of important inhibitors.

We very much thank the reviewer for these positive comments.

Whereas, globally well written, the document needs important improvements and clarifications:

1) Based on their observation that *bc1-aa3* mutant is only partially attenuated in mice, authors suggest that inhibition of *bc1* by a drug may not improve human chemotherapy: First, the observation that *bc1* inhibitors are only bacteriostatic and poorly effective in mice has been largely documented already. Here, authors wanted to demonstrate their strong and important assumption using the marmoset model. However, it is important to be cautious with the obtained data, as the treatment of marmoset by ND-10885 may not be fully comparable to the treatment with mechanistically related compounds such as Q203 (or arylvinylpiperazine amides).

We agree with the reviewer that lack of bioavailability precluded us from making conclusions about efficacy of Q203 in marmosets. Clearly, we need to strike a balance between raising caution due to the alarming finding that ND-10885 increased cavitation in marmosets - which should be considered when *bc₁-aa₃* inhibitors are evaluated in humans - and the fact that we can't make conclusion about the effect of other *bc₁-aa₃* inhibitors besides ND-10885 in marmosets. We feel that our manuscript achieves this balance.

In regard to previous studies on Q203: First, Pethe et al (reference #10) reported that Q203 alone is sufficient to kill H37Rv in mice (Fig. 2, PMID 23913123). Then Kalia et al (reference #21) reported Q203 to have no effect on H37Rv in mice (Fig 6D, PMID 28652330). Most recently, Foo et al (reference #12) reported static activity of Q203 in a mouse model of acute TB. (Fig 5 of PMID 30301850). Three reports, three fundamentally different claims for the impact of Q203 on *Mtb* in mice ranging from cidal, to stasis, to being inactive. We thus believe that there is significant value in demonstrating the impact that complete inactivation of cytochrome *bc₁-aa₃* oxidase inactivation (as achieved by genetic inactivation) has on *Mtb* in mice.

2) Authors have chosen to switch from Q203 to ND-10885 because of PK and exposure problems. The choice of replacing Q203 by this compound has some possible consequences: -)As shown on Sup fig 10, ND-10885 is far from being an ideal compound in term of stability in vivo. ND-10885 is shown prone to metabolic transformation by liver enzymes. Authors should add in the text (line 118-120) a link to the supplementary fig. 10 attracting attention on ND-10885 metabolism.

This has been added (see line 129). Supplementary fig. 10 is now supplementary fig. 13.

3) Part of the argumentation sits on the assumption that ND-10885 and Q203 share the same mode of action. However, it is very surprising that ND10885 has not been tested, with the long list of other compounds shown in table S1 and S2, against the various ETC mutants. It is essential to prove that ND10885 shows the same profile than Q203 against the various mutants, in particular an absence of effect against the Δ ctcA-qcrCAB mutant.

We respectfully disagree with the reviewer because ND-10885 has been well defined as an inhibitor of cytochrome *bc₁-aa₃* oxidase in previous studies. ND-10885 has the same phenotype as Q203 in that the laboratory adapted strain H37Rv overcomes growth inhibition within 2 weeks of exposure despite the fact that reduction of Alamar Blue is inhibited, the *cydC::aph* mutant is hyper-susceptible to this compound whereas QcrB mutants with defined amino acid substitutions known to confer cross-resistance to Q203, also confer cross-resistance to ND-10885.

Strain	ND-10885 MIC (μ M)	Q203 MIC (μ M)
H37RV	>50	>5
Δ cydC	0.039	<0.0024
Δ cydC::qcrB M342I	0.78	0.02
Δ cydC::qcrB A317T	1.56	0.06
Δ cydC::qcrB M342T	1.20	0.06
Δ cydC::qcrB 312G	4.7	1.25
Δ cydC::qcrB S182P	3.13	0.03
Δ cydC::qcrB A317V	12.5	2.5
Δ cydC::qcrB A396T	3.13	0.03

4) The conclusions regarding the efficacy of ND10885 in animals must be exclusively restricted to this compound. Extrapolating these observations to other compounds targeting the same pathway should be cautiously discussed.

We believe that our discussion is as cautious as it should be. The most relevant section is in lines 246-268. We make the following general conclusion: *inactivation of Mtb's cytochrome bc₁-aa₃*

*oxidase alone is insufficient to kill Mtb and **could possibly** complicate treatment by increasing cavity formation.* We believe that this statement is justified. All other points we make in this section are specifically for ND-10885. Nevertheless, in response to this criticism we added another sentence to the discussion pointing out that we are not making conclusions for inhibitors other than ND-10885 (lines 266-268).

5) As this is the first time that a group relay a double KO Ndh-2 (Ndh+NdhA), which was suggested impossible by others (Vilchèze, PNAS2018), it would be important to confirm the genotype by WGS to complete Southern blot data. In the same way, southern blots reporting on the deletion of bc1-aa3 are showing high background signal, which makes difficult to be formal on the deletions. In addition, the EcoR1-EcoR1 band observed in the Southern blot of the Δ ctaD clone is bigger than 1.5 kb, which do not match with the expected 1360 bp. As this part of the work is central in the manuscript, I would highly recommend to verify all mutants by WGS.

We agree with the reviewer in the fact that these are mutants long deemed to be not viable by the TB research community and that we should confirm their genetic identity beyond doubt. Following the reviewer's suggestion, we have confirmed the mutant's genotype through WGS, and we have substituted the Southern blot data by WGS reads alignment (Supplementary Fig. 2 and Supplementary Fig. 8).

6) Line 86. There is 1.5 log difference between WT and Δ ctaC at day 110. I don't think one can write "similar titers" in this situation. Nonetheless, at the view of the progression of the curve of the Δ ctaC mutant, it is difficult to predict whether equivalent titers would be reached few weeks later. In addition, in Fig1, Δ ctaC-comp shows growth equivalent to the one of Δ ctaC. Thus, it does not seem correct to say that "The final CFU titers were lower for Δ ctaE-qcrCAB and Δ ctaC than for WT Mtb, which suggest that the cytochrome bc1-aa3 oxidase is indeed required for full virulence of *M. tuberculosis*". In addition, this sentence is contradictory with the previous one, claiming : "SIMILAR titers" between WT and mutant Δ ctaC.

We accept this criticism. The manuscript was edited to make the description of this data clearer (see the paragraph beginning in line 89).

7) It would be good to propose some hypothesis to explain the differences of "toxicity" of long lipids versus cholesterol for the NDH-2 mutant.

Our explanation for this difference is the following: cholesterol only undergoes 3 cycles of β -oxidation (in the side chain), while, for example, to fully degrade oleic acid it is necessary 9 β -oxidation cycles. Hence, since our data strongly suggests that NADH generated during β -oxidation is implicated in the phenotype, we think that cholesterol degradation does not lead to the formation of enough NADH to reach a toxic effect. We have added this hypothesis to the discussion (see the paragraph starting in line 280).

8) Some claims of novelty should be reformulated. For example "We conclude that Mtb requires the cytochrome bc1-aa3 supercomplex to achieve optimal growth rates and maximal titers, but that this complex is not essential for growth or persistence a long as Mtb can express a functional cytochrome bd oxidase". This fact has been previously reported by many others: for example, Kalia et al, or Foo et al , showed that the inhibition of bc1:aa3 by Q203

is not able to fully block the electron flow through the cytosol due to the presence of the cytochrome oxidase bd. Same authors reported “multiple lesions and inflamed foci were found in the lungs of the mice (...) treated by Q203”. This raise the question of the novelty of some important data presented as novel in this manuscript such as “These findings raise concerns about the use of cytochrome bc₁-aa₃ oxidase inhibitors to treat TB.”

We recognize that indeed concerns regarding the use of cytochrome bc₁-aa₃ oxidase inhibitors to treat TB were stated before (PMID: 28652330), and we have edited the manuscript to address this point (see lines 115-116). But please also consider our response to point 1.

9) Authors show that NDH-2 mutants are only slightly attenuated in mice (fig 4D-E). Due to the very different type of growth, metabolisms and lesions, and thus carbon source in mice versus marmosets, and the relationship shown here between lethality of Δ NDH-2 and lipids, it would be much more relevant to test NDH-2 mutants in marmosets.

Certainly, this would make a more definitive case for Δ ndh-2 *in vivo* dispensability. However, additional studies in marmosets, which are a major undertaking are simply beyond the scope of this report.

Minor points:

1) The chemical structure of ND-10885 should be shown in the main text, also with Q203, to show that these two compounds have substantial dissimilarities.

These structures have been published previously, but we included Supplementary Figure 15 showing the conserved imidazo[1,2-a]pyridine-3-carboxamide core in both ND-10885 and Q203.

2) Fig.S1. Put the mean of the experiments instead of “one representative of at least two”.

Due to experimental constraints, samples were taken at different time points in the two biological replicates, so we cannot calculate mean ODs. Nevertheless, both biological replicates showed the same growth profile.

3) Fig2C and D could be transferred in supplementary. Their reading is quite challenging and it is almost impossible for the reader to make its own general conclusion out of it.

We prefer to keep these figures in the main text and hope that the reviewer can agree that this was a minor criticism.

4) Fig5D: not needed

The figure was deleted

5) Fig5E can be transferred in supplementary data

Fig. 5E is now Supplementary figure 10.

6) Line 215: structures of DDD00853663, DDD00946831 should be included in sup. data

This has now been included in Supplementary Figure 14.

7) FigS6. The probe is missing in part B (WT)

As stated above, we have substituted the Southern Blot data in Supplementary Fig. 8 for WGS reads alignments.

8) Line 120: add sup fig. 10

The manuscript was edited accordingly (see line 129).

9) Not enough data (supplementary) are given regarding genetic constructions leading to gene deletions. Experiments should be sufficiently detailed to be reproducible by colleagues.

We have extended the material and methods section regarding mutant construction in order to make this point clearer and more easily reproducible by others (see lines 351-394). Lists of strains and plasmids were also added as Supplementary Table 5 and 6.

Reviewer #3

In the current study the authors have tried to address the non-redundancy of Mtb respiratory chain components and tried to elucidate the associated non-essentiality/dispensability of these components for the survival of Mtb. The significance of the studies conducted here is very high and the paper is well written. Also, considering that drug Q203 is in phase 2 clinical trial, this data serves best fit pre-clinical data integrating molecular biology, mouse and nonhuman primate studies. In general I felt that there could have been a better discussion of future directions at the end of the manuscript. Overall a manuscript that after revisions will be highly improved and worthy.

While the authors have done commendable amount of work, the manuscript however has some issues and shortcomings which I will try to point out:

We thank the reviewer for both the positive remarks and the constructive criticism.

Major issues:

1. Line 47: While the authors briefly mention a previous study, which reported M.tb lacking both D-ndh/ndhA to be not viable, they do not address the finding in a convincing way. Neither are the associated results and molecular methodology explained in enough details to support their contrasting claim, while the cited study does describe their methods and approaches in details enough to make the reader confident in the reported results.

We thank the reviewer for raising this point. We have extended the material and methods section dedicated to mutant construction in order to be more detailed (see lines 351-394).

We have also sequenced the genome of the $\Delta ndh-2$ mutant to confirm its genetic identity. Southern Blot data were substituted by WGS reads alignment in Supplementary Fig 8.

2. Line 58: While the study on clofazimine does interrogate activation by NDH-2, it doesn't rule out the non-redundancy of this reaction by other oxidoreductases which needs to be mentioned in this context.

We agree that other oxidoreductases might reduce clofazimine. This is referred in lines 298-300.

3. The animal experiment lacks mention of statistical method involved and the extent of significance, and is thus not rigorous.

We have tested the statistical significance in each time point and updated the figures related with mouse infections (Fig. 1, Fig. 4, Supplementary Fig. 3 and Supplementary Fig. 6).

4. Fig 2 & 3: All marmoset data lacks infected controls. Additionally, Figure 2 mentions *Klebsiella* superinfection? Furthermore, Fig 2A shows significant difference in only 2 lesions and Fig 2B in only 3. The inferences drawn are non-convincing and not strong.

Infected untreated control animals have been previously published (PMID: 25941223, 23716617) and animals would not have survived beyond 8 weeks in the absence of treatment. All of the lesions show significant declines in PET SUV after the initiation of treatment except one. The response to treatment with ND should be compared to the response seen in Figure 2 of 25941223 which shows the more typical drug impact of consistent reduction of both SUV and disease volume. The inconsistency of this response is what is remarkable about ND10085, it does stop disease progression in the sense that the animals do not develop new lesions nor do their existing lesions continue to grow, instead the pathology of the lesions change. We infer that this change is due to progression of hypoxic lesions within caseum that are not respiring on oxygen and are therefore insensitive to QcrB inhibition.

5. Fig 3A: Historical data for different regimens appear to have completely different dynamics and therefore makes the comparison inappropriate. Also, curiously 2-drug streptomycin-isoniazid (HS) regimen outperforms the 4-drug isoniazid-rifampicin-pyrazinamide-ethambutol (HRZE) regimen.

The published data do have different dynamics, that's the whole point, the large shift to primarily cavitory lesions and replication of the bacteria to high titer is unprecedented but completely consistent with what we know about the pathophysiology of caseous lesions and their progression to cavities. The two drug regimen performs significantly worse as reported in the publication from which these animals were described (PMID: 25941223). This figure reports only the pathology of the lesions post-treatment which we did not previously describe to show the impact of QcrB inhibition.

6. In comparing the images from the current manuscript with a previous paper from some of these authors published in collaboration with Dr. Joanne Flynn, it appears that previously distinct mediastinal lymph node involvement was observed in *Mtb* infected animals, whereas those receiving Q203 don't appear to have any. Although *Mtb* CFUs delivered were listed in the previous paper and are not described in the current study so it could be a function of that but if not then this drug could be a game changer for treatment of EPTB, and perhaps the authors could discuss that? The authors recognize that that bd oxidase is dependent on the oxygen tension and is strongly upregulated during hypoxia. The PET CT scan mentions only about lung lesions where the oxygen tension is high. Analysis of lesion characteristics in

relatively anaerobic tissue like mediastinal or cervical lymphnodes will further clarify efficacy of this drug in extrapulmonary TB conditions which are relatively difficult to manage with currently prescribed drug regimens.

TB infected marmosets do get extrapulmonary disease but it depends on the strain used to infect them. We would have to do another round of infection with a more virulent strain to have enough data to report on activity against infected lymph nodes. The caseous center of lesions in these animals has been shown to be hypoxic in numerous prior studies and the progression of these lesions in the animals suggests limited scope for activity in EPTB.

7. Though the authors intend to compare the current study with their previous study involving 4 drug chemotherapy, starting the therapy 6 weeks after infection suggests that it was too late and too less in this case. A previous study by (Cadena et.al.) suggest that even with low dose infection with 7 CFU CDC1551 cavitory lesions were reported at 4 weeks, initiating the therapy early could have seen better results.

Early treatment of TB patients is of course desirable, but frequently impossible. We therefore designed the model to ask what would be the impact in patients presenting with advanced active disease. We believe that this results in relevant and realistic results.

8. Line 110 : “Treatment of Mtb-infected marmosets with ND-10885 controlled infection and inflammation, but increased the occurrence of cavitory lesions” is misleading. As shown in fig 3. No. of cavitory lesions and caseous necrosis has only increased in the ND-10885 group. It is insufficient to conclude on infection and inflammation status only on basis PET-CT findings of lung lesion glycolysis and hard volume, adding biochemical data like CRP or other inflammatory markers would help corroborate the inflammation status better.

CRP is neither very sensitive nor very specific. Our conclusions are a simple description of the data, the infection was controlled (as evidenced by the fact that the animals did not all die), inflammation was decreased (as evidenced by the FDG signal diminishing) and cavitory disease increased (as evidenced both by the CT apparent cavities and by gross pathology at the end of treatment). Adding CRP or other inflammatory markers would have no value to assessing the impact of this drug.

9. Survival in control group of previous study (Via LE) was around 9-10 weeks whereas according to Fig 2.a.) and 2b.) survival seems to be 111 days. This shows that the treatment drug clearly has superior survival benefits than control group. A survival analysis curve involving all the animals in fig 3. will give a better representation in this context.

It is stated in the manuscript that ND-10885 arrests disease progression (e.g. lines 135-136) and does promote survival, but it does so at the price of allowing pathology to advance. We added a repeat of this statement in the discussion (lines 255-256).

Minor issues:

1. The methods are not well described and need substantial improvement to make the results appear convincing. The molecular strategy needs to be elaborated in detail and include accurate mentions of plasmids, primers and validations of successful interventions. Furthermore, the should also expand on the methodology employed for estimating CFU and

growth rates throughout the study. Sticking to a general presenting style will make the data appear more understandable (preferably log scale which is customary of the field for *in vivo* animal pathogenic burden and CFU/ML for *in vitro*).

We have given more detail on the material and methods regarding mutant construction and we added Supplementary Tables 5 and 6 with all the strains and plasmids used in this work. The *in vivo* CFU data in this manuscript are in log scale. We updated the figures to make this more obvious.

2. The introduction is disparate and not continuous. The published research in the field need to be mentioned and cited in context of current rationale. Most of the citations need to be revised for accuracy.

We have edited the introduction and revised citations to make it more consistent.

3. Line 68-70: Citations and prior reports suggesting NDH-1 and its probable compensation for NDH-2 needs to be mentioned and discussed.

This sentence refers to the reasoning we have used to start this work, i.e if *Mtb* has an alternative enzyme, why is NDH-2 essential? As far as we know, there are no other reports demonstrating that NDH-1 can effectively compensate NDH-2. We did add references showing that *Mtb* encodes an NDH-1.

4. Results: Fig 1A & B: Why does the complemented strains appear to outperform even wild type in murine model? Were the strains validated for confirming they are still not merodiploid? Were the cloned enzymes being overexpressed? If not, why does the complemented strain has a survival advantage? Also, why the same strain has a slightly diminished growth than WT *in vitro* Suppl. Fig 1D??

Mtb's respiratory chain is a highly plastic and adaptive process, and thus it is hard to predict the phenotypic consequences of a differential *ctaE-qcrCAB* expression between the wild type and complemented strain. What we can say is that re-introducing a copy of the operon *ctaE-qcrCAB* rescued the knockout's *in vivo* phenotype, and partially recued the *in vitro* phenotype, strongly suggesting that the slow growing phenotype is due to a non-functional cytochrome *bc1-aa3* oxidase.

Fig 1C&D: It is surprising to see that the Δ ctaC strain is diminished for growth. This should be discussed. If there is any data on the *in vivo* burdens of Δ ctaD-qcrCAB or Δ ctaC-qcrCAB or Δ qcrCAB, the authors are encouraged to show that. Alternatively this data could be moved to the supplementary material.

Our main point was to show that both mutants with a non-functional terminal oxidase were still able to grow and survive *in vivo*. Nevertheless, we do not have enough knowledge about the respiratory chain, nor we have data to pinpoint the reasons that lie behind the observed differences between Δ ctaC and Δ ctaE-qcrCAB. As suggested by the reviewer we have made Δ ctaC mouse infection data the new Supplementary Fig. 3.

6. Suppl Fig 4: This figure depicts one of the most convincing findings of this study and clearly establishes the compensatory roles of cytochrome bd oxidase and cytochrome bc1-aa3 complex for in vivo survival in mice model. Statistics needs to be added though.

A statistical analysis was performed, and Supplementary Fig. 6 (previously Suppl. Fig. 4) was updated accordingly.

7. Line 123: What was estimated infection dose in CFU for the low dose aerosol infection in marmosets? This detail should be provided.

This has been published previously; we generate an aerosol titer on mice to have 250CFU/L and then expose animals to that aerosol for a delivered dose of 10-25 CFU per animal. This information has been added to the manuscript (lines 487-488).

8. Fig 3C: It would be best if quantitative analysis of multiple fields was performed.

We don't have the ability to do this quantitatively at the moment, this will be reported in future analyses, this was typical of multiple lesions examined (at least two from each animal).

9. Line 368-371: Whether the complementation was done under the control of self-promoter or any constitutively expressing strong (hsp60/myc) promoter?

This information was added to the mutant construction section.

10. There is no information about the dose used for infection. This is an important gap that should be filled

See response to #7.

11. Fig3: no control data with infected untreated (3A). The "HS" seems to work better than HRZE treatment. Can the authors present any suitable explanation. Fig3B should be supplemented by an infected untreated control.

Same as comment to criticism #9 (major issues) above.

12. Line 117-118: is fundamental information about the activity of the drug ND-10885 against Mtb? E.g., MIC?

We have included the MIC value for CDC1551, the Mtb strain used for marmoset infection. See also Supplementary Table 2.

13. Line 47: A reference could be added on the study that showed that ndh was dispensable for growth of Mtb.

The reference is in line 45 (reference number 8)

14. In Supplementary Figure 1 A- H, only panels D and H have their scales different on the Y axis, it could be aesthetically better to have all the scales same.

Supplementary Fig. 1 was changed as requested.

15. The gel picture in Panel D in Supplementary figure 2 is of a very low quality.

We agree that the Southern Blot is not of the best quality. We have substituted the Southern blot data by WGS reads alignment for a more clear-cut confirmation of the genetic identity of these mutants.

16. Why weren't the mutants (TetOFF and deletion mutants) not studied for growth for the same number of days? TetOFF were monitored for growth up to 20 days while DctaE-qcrCAB were studied for 25 days and DcydABDC ctaE-qcrCAB-TetOFF was studied for more than 30 days. This should at the very least be described.

We have tried to be consistent in that we show growth curve until stationary phase, although the last time points are not the same. We have edited the manuscript accordingly.

17. Line 87 seems a little vague with the authors concluding that the lower CFU titres are responsible for the lower virulence of Mtb. The authors have not shown CFU data (all figures are presented in OD) and no experiments were conducted at this point to correlate the CFU with any virulence parameter to enable this conclusion.

This paragraph refers to Fig. 1A-D which presents data as CFU per lung. We have added the reference to Fig.1 in the end of the paragraph.

18. Lines 91 to 94 appear a little unnecessary in the results section, perhaps they should be moved to the discussion section.

We understand the point raised by the reviewer, but we feel that it is important to give some context to the terminal oxidases synthetic lethality data, given that there are other reports pointing in the same direction.

20. Supplementary Figure S4. The scales are not consistent on the panels (3 out of 4 have 125 days while A has 80 days on the X axis and two have 10^6 while 2 have 10^7 on the Y axis).

Thank you for pointing this out; we have made the panels consistent regarding scales on both x and Y axes.

21. Line 116: What could be the reason for the Q203 not being able to provide acceptable exposure in marmosets?

Q203 is extremely insoluble, previous studies in rodents have used DMSO to solubilize and dose this, this is not allowable in NHP, we tried a wide variety of excipients and failed to find any acceptable way to dose Q203.

22. Figure 3C: The pathology panel has some factors that needs improvement. The first picture on top left has a small box that needs to be connected with some arrows or lines to the bottom enlarged picture to easily demonstrate that the bottom picture is an enlargement of that small box. As it stands now, unless someone reads the legend, will think these are different pictures.

The figure has been modified as requested.

23. The contrast of the figure on the bottom needs to be improved as right now it is difficult to visualize the marked regions clearly.

We spent hours trying to get better contrast, it is not possible – acid fast staining in tissue is very difficult. We expect that a higher resolution will improve this figure.

REVIEWERS' COMMENTS:

Reviewer #1 (Remarks to the Author):

I have been through all the responses of the authors and the new data they have provided. All my concerns have been addressed. This is an outstanding piece of work and I look forward to seeing it published.

Reviewer #2 (Remarks to the Author):

This reviewer is satisfied by the answers and the modifications proposed by the authors.

Reviewer #3 (Remarks to the Author):

The revised manuscript is sufficiently receptive to prior critique and is improved as a result.